# Direct OC-CHO coupling towards highly $C_{2+}$ products selective electroreduction over stable $Cu^0/Cu^{2+}$ interface

Xin Yu Zhang[1,6], Zhen Xin Lou[1,6], Jiacheng Chen[2], Yuanwei Liu[1], Xuefeng Wu[1], Jia Yue Zhao[1], Hai Yang Yuan[1] ✉, Minghui Zhu[2], Sheng Dai[3], Hai Feng Wang[4], Chenghua Sun[5], Peng Fei Liu[1] ✉ & Hua Gui Yang[1] ✉

Electroreduction of $CO_2$ to valuable multicarbon ($C_{2+}$) products is a highly attractive way to utilize and divert emitted $CO_2$. However, a major fraction of $C_{2+}$ selectivity is confined to less than 90% by the difficulty of coupling C-C bonds efficiently. Herein, we identify the stable $Cu^0/Cu^{2+}$ interfaces derived from copper phosphate-based (CuPO) electrocatalysts, which can facilitate $C_{2+}$ production with a low-energy pathway of OC-CHO coupling verified by in situ spectra studies and theoretical calculations. The CuPO precatalyst shows a high Faradaic efficiency (FE) of 69.7% towards $C_2H_4$ in an H-cell, and exhibits a significant $FE_{C2+}$ of 90.9% under industrially relevant current density ($j = -350$ mA cm$^{-2}$) in a flow cell configuration. The stable $Cu^0/Cu^{2+}$ interface breaks new ground for the structural design of electrocatalysts and the construction of synergistic active sites to improve the activity and selectivity of valuable $C_{2+}$ products.

The electrosynthesis of multicarbon ($C_{2+}$) products such as ethanol ($C_2H_5OH$) and ethylene ($C_2H_4$) from $CO_2$ is highly attractive because of the versatility of these products in the chemical and energy industries[1–4]. However, the selective production of $C_{2+}$ products from $CO_2$ reduction reaction ($CO_2RR$) is challenging, with competition from the hydrogen evolution reaction (HER) and $C_1$ products (e.g., CO, HCOOH, and $CH_4$) production[5–7]. Among all the nanostructured electrocatalysts reported thus far, copper-based materials are known to be the most selective for $CO_2$-to-$C_{2+}$ production[8,9]. Up to now, a variety of strategies have been proposed to improve $CO_2$-to-$C_{2+}$ selectivity, including controlling oxidation states[10,11], constructing nanostructures[12,13], alloying[14,15], doping[16], and molecular decorating[1].

Unfortunately, the selectivity of $C_{2+}$ products is still low, especially at high current densities ($j$)[17].

Recently, manipulating oxidation states for constructing a synergistic $Cu^0/Cu^{1+}$ interface has been demonstrated as an effective way to promote $C_{2+}$ product conversion via stabilizing $Cu^{1+}$ species[11,12,17–22]. For example, it's reported that a high concentration of $Cu^{1+}$ (8%) in the iodine-modified Cu catalyst was observed by the linear combination fitting (LCF) analysis of operando X-ray absorption spectra (XAS), which contributed to the high $FE_{C2+}$ of about 80% at the $j$ lower than 40 mA cm$^{-2}$ (ref. 12). Our group also constructed the stable $Cu^0/Cu^{1+}$ interface via isolating Cu-S motif with $C_{2+}$ selectivity over 80%[22]. By combining $Cu^0$ and $Cu^{1+}$ synergistic species, the adsorption

[1]Key Laboratory for Ultrafine Materials of Ministry of Education, Shanghai Engineering Research Center of Hierarchical Nanomaterials, School of Materials Science and Engineering, East China University of Science and Technology, 130 Meilong Road, Shanghai 200237, China. [2]State Key Laboratory of Chemical Engineering, School of Chemical Engineering, East China University of Science and Technology, 130 Meilong Road, Shanghai 200237, China. [3]Key Laboratory for Advanced Materials and Feringa Nobel Prize Scientist Joint Research Center, Institute of Fine Chemicals, School of Chemistry and Molecular Engineering, East China University of Science and Technology, 130 Meilong Road, Shanghai 200237, China. [4]Key Laboratory for Advanced Materials, Centre for Computational Chemistry and Research Institute of Industrial Catalysis, School of Chemistry and Molecular Engineering, East China University of Science and Technology, 130 Meilong Road, Shanghai 200237, China. [5]Department of Chemistry and Biotechnology, and Center for Translational Atomaterials, Swinburne University of Technology, Hawthorn, VIC 3122, Australia. [6]These authors contributed equally: Xin Yu Zhang, Zhen Xin Lou. ✉e-mail: hyyuan@ecust.edu.cn; pfliu@ecust.edu.cn; hgyang@ecust.edu.cn

of *CO can take place at different active sites, resulting in the improvement of CO dimerization, i.e., the $Cu^0$ site can activate $CO_2$ and facilitate the following electron transfers, while the $Cu^{1+}$ site strengthens the *CO adsorption and boosts C-C coupling, further endowing a lower Gibbs free energy for *OCCOH formation[11,21,23–25]. Similarly, it was revealed that over the two adjacent $Cu^0$ and positively charged $Cu^{\delta+}$ atoms, the $Cu^0$ site can adsorb $CO_2$ and the neighboring $Cu^{\delta+}$ site is conducive to $H_2O$ adsorption, thus promoting the activation of $CO_2$[26]. Moreover, theoretical studies have shown that the subsurface O stabilized surface $Cu^{\delta+}$ species indeed enhances the $C_2$ products selectivity through increasing the *CO coverage[27].

In addition to the widely reported $Cu^{1+}$ species in the field of $CO_2RR$, $Cu^{2+}$ species with higher oxidation states than $Cu^0$ or $Cu^{1+}$ sites feature the structure characteristics to easily bind CO or $H_2O$, which have been demonstrated in other heterogenous catalysis fields[28–30]. Recently, Qiao et al. have proved the feasibility of $Cu^{2+}$ sites for promoting *CO hydrogenation in the Cu-Ce-$O_x$ solid solutions, which facilitate the generation of *CHO intermediates instead of *CO dimerization on the single $Cu^{2+}$ sites[31]. As a consequence, Cu-Ce-$O_x$ delivered a high selectivity for $CH_4$ with significant suppression of the $C_2$ products. The above findings suggest the possibility of retaining $Cu^{2+}$ sites through material structure design and promoting *CHO intermediate formation on $Cu^{2+}$ catalytic active sites. Inspired by these works, we anticipate that the construction of $Cu^0$/$Cu^{2+}$ interface may provide the possibility of direct OC−CHO coupling process compared with single $Cu^{2+}$ sites; however, it still remains a grand challenge to build stable $Cu^0$/$Cu^{2+}$ interface under operando $CO_2RR$ conditions.

Herein, we rationally screened the Cu-based precatalysts for constructing stable $Cu^0$/$Cu^{2+}$ interfaces, and elucidated their synergic roles in $CO_2$-to-$C_{2+}$ conversion. Using Materials Project Database (MPD) and density functional theoretical (DFT) calculations therewith, we screened different $Cu^{2+}$-containing compounds and found $Cu^{2+}$ phosphorus oxysalts of $Cu_2P_2O_7$ and $Cu_3(PO_4)_2$ (collectively named CuPO) to be the most promising candidate for exhibiting the highest stability of $Cu^0$/$Cu^{2+}$ interface under the electroreduction condition; in consequence, the peculiar $Cu^0$/$Cu^{2+}$ interfaces facilitate a low-energy pathway of *CHO coupling with *CO to form *OCCHO intermediate. Experimentally, we synthesized CuPO catalysts through a facile and scalable method, which achieved 69.7% FE for $C_2H_4$ in the neutral electrolyte, and 90.9% $FE_{C_{2+}}$ with a $C_{2+}$ partial current density ($j_{C_{2+}}$) of over 300 mA cm$^{-2}$ in a flow cell using the alkaline electrolyte. Our operando experimental characterizations unambiguously demonstrate the robust existence of $Cu^0$/$Cu^{2+}$ interfaces derived from CuPO during $CO_2RR$, and in situ surface-enhanced infrared absorption spectroscopy (SEIRAS) in conjunction with DFT calculation testify that the $Cu^{2+}$ sites are conducive to the formation of *CHO, which then facilely coupled with *CO on $Cu^0$ surface to form *OCCHO intermediate, leading to high-efficiency $CO_2$-to-$C_{2+}$ conversion performance.

## Results
### Theoretical calculation
The formations of *CO, *COH, and *CHO intermediates on $Cu^0$, $Cu^{1+}$ and $Cu^{2+}$ sites induce different coupling manners and probabilities for $C_{2+}$ species formation on $Cu^0$/$Cu^{1+}$ and $Cu^0$/$Cu^{2+}$, which provide guidance on designing the Cu-based catalysts for $CO_2RR$ to $C_{2+}$ products[11,31]. Firstly, we explored the formation energy of *CHO or *COH (*CO + H$^+$/e → *CHO or *COH) on classical $Cu^0$ (metallic Cu), $Cu^{1+}$ ($Cu_2O$), and $Cu^{2+}$ (CuO) sites, respectively, which play key roles in the $C_{2+}$ products. From Fig. 1a, one can see that on the $Cu^0$ and $Cu^{1+}$ sites, the *CHO or *COH formation is relatively endothermic at the potential $U = 0$ V (all potentials were calibrated to the reversible hydrogen electrode (RHE) if not mentioned), indicating that *CO could be the primary intermediates. By comparison, the $Cu^{2+}$ site has an excellent

ability to hydrogenate *CO to form *CHO ($\Delta G = 0.10$ eV) rather than *COH ($\Delta G = 1.33$ eV). Accordingly, it can be expected that the $Cu^0$ and $Cu^{1+}$ sites could be covered by CO intermediates, while the $Cu^{2+}$ site contributes to *CO hydrogenation to form *CHO intermediates that can facilitate the direct OC−CHO coupling process to increase the selectivity of $C_{2+}$ product (Fig. 1b).

Facing various Cu-based catalysts with $Cu^{2+}$ sites, it is difficult and time-consuming to examine each one and locate the optimal one with superior electrochemical stability and catalytic performance using complex experimental methods. High-throughput screening as an efficient forecast method for a large group of candidate materials has been widely used to predict promising materials for experimental synthesis. Here, we extracted 83 $Cu^{2+}$-containing compounds from Materials Project, and various criteria were utilized to filtrate candidates (details in the Supplementary Note 1). Firstly, the thermodynamic stability that is the intrinsic property of materials was considered, in which two criteria were used: (i) the formation energy ($E_f$) should be less than 0 eV, and the more negative $E_f$ means the higher thermodynamic stability; (ii) the energy above the convex hull ($E_{hull}$) should be less than 70 mV/atom[32], which can further quantify the structural stability of materials, and the smaller $E_{hull}$ indicates that the corresponding material is more stable. Accordingly, 26 candidates were screened out (Fig. 1c). Then, their electrochemical dissolution potential $U_{diss}$ was further assessed (see details in the Supplementary Note 2), and a more positive $U_{diss}$ means higher electrochemical stability. As shown in Fig. 1d, it can be found that 9 $Cu^{2+}$-containing compounds exhibit good electrochemical stability, in which $Cu_2P_2O_7$ ($U_{diss} = 0.94$ V) and $Cu_3(PO_4)_2$ ($U_{diss} = 0.77$ V) process the optimal electrochemical stability. Meanwhile, the high formation energy of the $O_{vac}$ on $Cu_3(PO_4)_2$ (0.82 eV) indicated that the phosphate group could play an important role in stabilizing the O atoms, thus limiting the reduction of $Cu^{2+}$ (Supplementary Fig. 3). Therefore, $Cu_2P_2O_7$ and $Cu_3(PO_4)_2$ could be two promising candidates to construct $Cu^0$/$Cu^{2+}$ interface for $CO_2RR$ to $C_{2+}$ products under the suitable reaction conditions.

In order to clarify the performance of the $Cu^0$/$Cu^{2+}$ interface in catalyzing $CO_2RR$ to produce $C_{2+}$ products, we selected $Cu_2P_2O_7$ and $Cu_3(PO_4)_2$ as examples to construct the $Cu^0$/$Cu^{2+}$ interface (Supplementary Fig. 1), and explored its ability to promote the C−C bond coupling in comparison with the common $Cu^0$/$Cu^{1+}$ interface and the classical Cu(100) surface (Fig. 1e, f). Here, we considered all three general collaborative pathways: (i) *CO + $^\#$CO → OC−CO, (ii) *CO + $^\#$COH → OC−COH and (iii) *CO + $^\#$CHO → OC−CHO (* and $^\#$ represent $Cu^0$ and $Cu^{2(1)+}$ sites, respectively)[8], and calculated their reaction energy energies, aiming to understand the C−C bond coupling mechanism in depth. As shown in Fig. 1e–g, $CO_2RR$ on the $Cu^0$/$Cu^{2+}$ interface always has a lower energy profile than that on the $Cu^0$/$Cu^{1+}$ one, regardless of the C-C bond coupling pathways, implying the better acceleration of the $Cu^0$/$Cu^{2+}$ interface for the C-C coupling. In addition, although the OC−COH coupling process has a relatively low barrier of 0.71 eV at the $Cu^0$/$Cu^{2+}$ interfaces compared with the OC-CO coupling process ($E_a = 0.95$ eV), COH formation is difficult and requires amount of energy, leading the high effective energy barrier (Fig. 1f). The formation of CHO is much easier compared with that of COH on $Cu^{2+}$ sites; importantly, the $^\#$CHO intermediate only needs to overcome a low-energy barrier of only 0.46 eV to couple with *CO at the $Cu^0$/$Cu^{2+}$ interface (Fig. 1g), which is also lower than the energy barrier (0.54 eV) for OCCO dimerization on classical Cu(100). These indicate that the OC−CHO coupling process at the $Cu^0$/$Cu^{2+}$ interface constructed by $Cu_2P_2O_7$ (or $Cu_3(PO_4)_2$, Supplementary Fig. 2) has an evident superiority over the OC−CO or OC−COH coupling processes, and at the same time, the $Cu^{2+}$ site can well facilitate the *CO hydrogenation to the CHO intermediate for high $C_{2+}$ yield instead of $C_1$ (Supplementary Fig. 4).

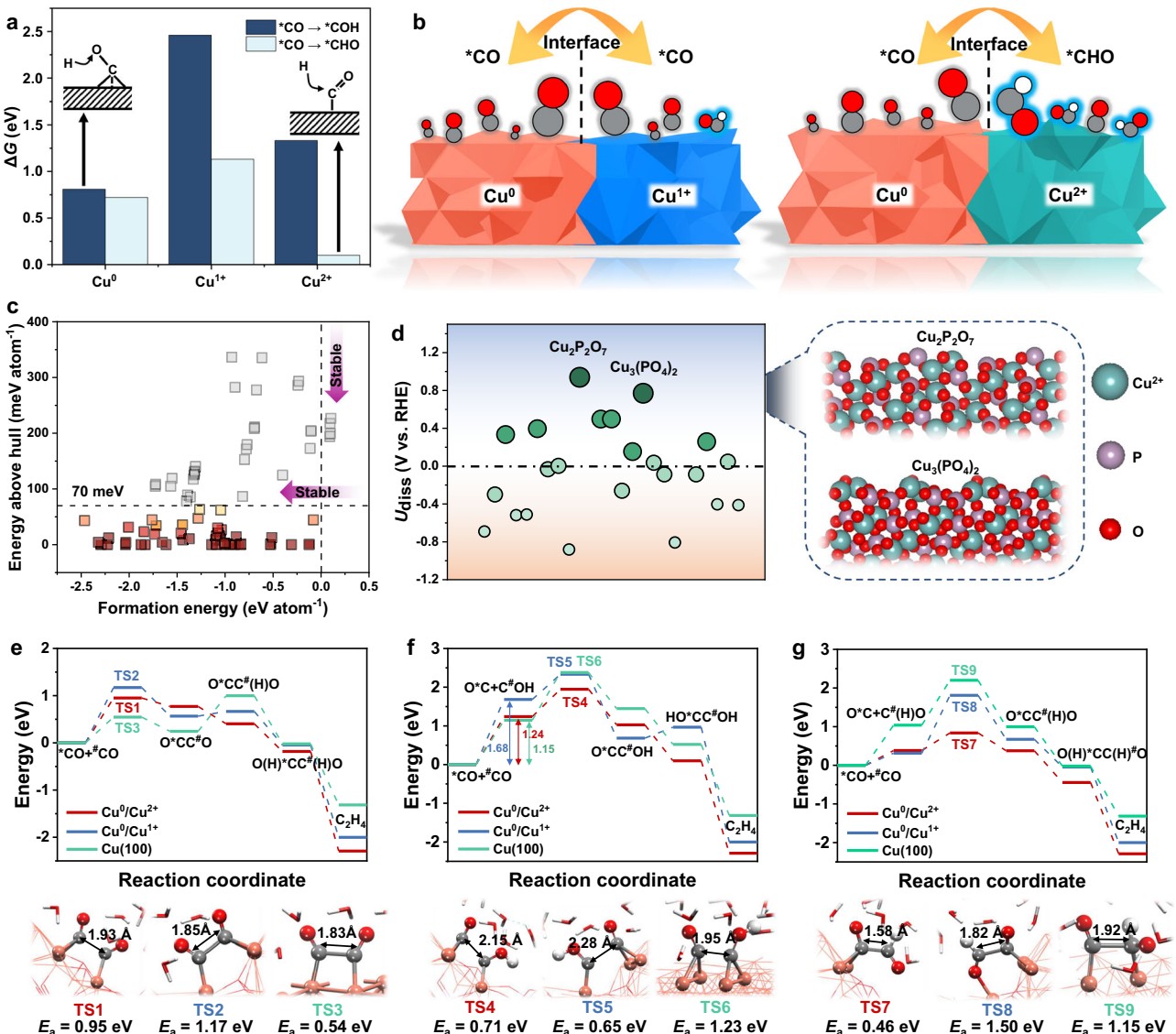

**Fig. 1 | Theoretical calculations. a** Reaction energy of *CO hydrogenation to *COH or *CHO on Cu, $Cu^{1+}$ and $Cu^{2+}$ sites, respectively. **b** Schematic diagram of different intermediates (CO or CHO) on the $Cu^0/Cu^{1+}$ and $Cu^0/Cu^{2+}$ interfaces. **c** Thermodynamic stabilities of materials as a function of $E_f$ and $E_{hull}$. **d** Electrochemical stability of candidates, where $U_{diss}$ is the dissolution potential of materials, larger and greener circles represent more stable materials (i.e., larger $U_{diss}$), and the structures of the most stable $Cu_2P_2O_7$ and $Cu_3(PO_4)_2$. **e–g** Energy profiles of different C–C coupling processes (* and # represent $Cu^0$ and $Cu^{2(1)+}$ sites, respectively), **e** CO-CO, **f** COH-CO, and **g** CHO-CO at the $Cu^0/Cu^{2(1)+}$ interfaces and Cu(100), and the related transition state structures (TS1 - TS9) of different C-C coupling processes.

## Catalysts synthesis and structure investigation

Based on the theoretical studies, CuPO catalysts were synthesized using a facile and scalable method, as illustrated in the "Methods", Supplementary Fig. 5 and Note 3. The XRD patterns of the catalyst powders investigated in this study are shown in Fig. 2a, b, with diffraction peaks well indexed to the triclinic phase of $Cu_3(PO_4)_2$ (PDF#97-006-8811) and the monoclinic $Cu_2P_2O_7$ (PDF#97-015-7107), respectively, without any impurities. After 1 h (h) of $CO_2RR$ at a potential of −1.40 V, both $Cu_3(PO_4)_2$ and $Cu_2P_2O_7$ are partially reduced to metallic Cu, indicating the coexistence of CuPO and metallic Cu components during $CO_2RR$. Furthermore, X-ray absorption fine structure (XAFS) analysis in conjunction with the XRD results proved the coexistence of $Cu^{2+}$ and $Cu^0$ species as the $CO_2RR$ goes on for at least 10 h (Supplementary Figs. 6 and 7). CuO as a control sample was synthesized using the same method, and XRD characterizations were also carried out to identify the material components (Supplementary Fig. 8). The shape and microstructure of $Cu_3(PO_4)_2$, $Cu_2P_2O_7$, and CuO

were examined via transmission electron microscopy (TEM), and all the three nanoparticles possess irregular shapes (Supplementary Figs. 9–11a). Supplementary Figs. 9–11b give the related high-resolution TEM (HRTEM) images of $Cu_3(PO_4)_2$, $Cu_2P_2O_7$, and CuO, and the clear lattice fringes with the spacing of 0.408, 0.316, and 0.252 nm are observed, corresponding to the (110) plane of triclinic phase $Cu_3(PO_4)_2$, the ($\bar{2}$02) plane of monoclinic phase $Cu_2P_2O_7$ and the (111) plane of monoclinic phase CuO, respectively. Further high-angle annular dark-field scanning transmission electron microscopy (HAADF-STEM, Supplementary Figs. 9c and 10c) and corresponding energy-dispersive spectroscopy (EDS) maps (Supplementary Fig. 9d–f and Supplementary Fig. 10d–f) verify that Cu, O, and P elements are homogeneously distributed along the $Cu_3(PO_4)_2$ and $Cu_2P_2O_7$ nanoparticles. Notably, after $CO_2RR$, the HAADF-STEM image of $Cu_3(PO_4)_2$ shows the agglomeration of small-size nanoparticles (Supplementary Fig. 12), and the HRTEM image exhibits that metallic Cu nanocrystal with a d-spacing of (111) plane is closely contacted to $Cu_3(PO_4)_2$ with a

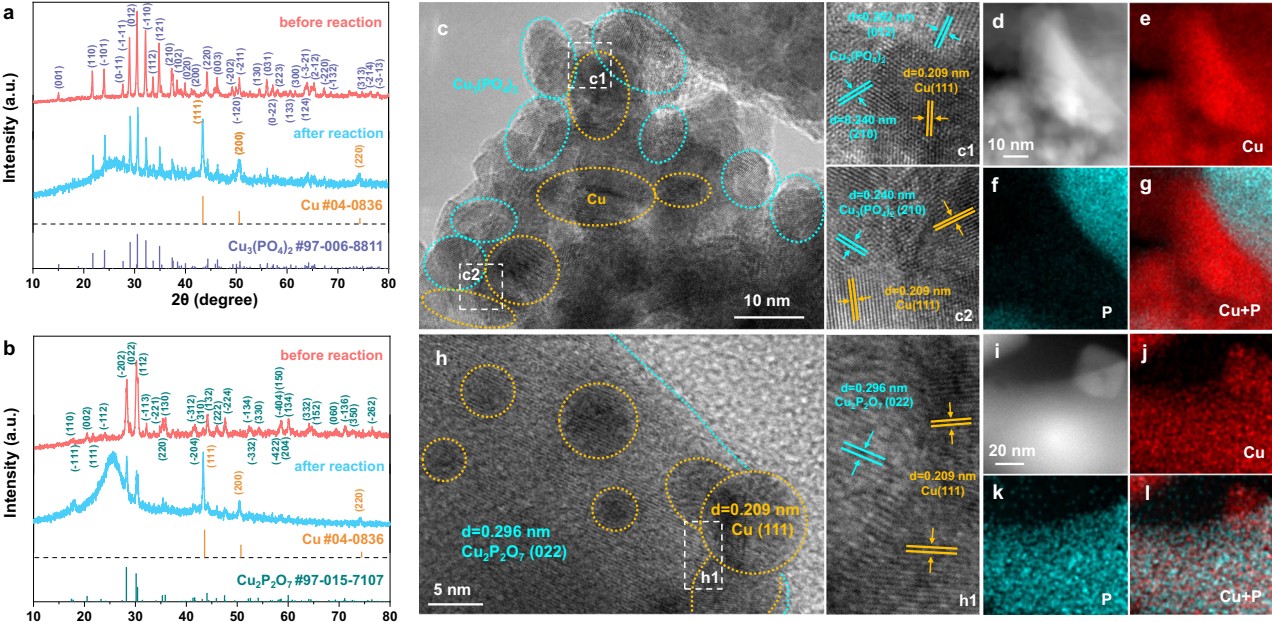

**Fig. 2 | Structural and compositional analyses. a, b** XRD patterns of **a** $Cu_3(PO_4)_2$ and **b** $Cu_2P_2O_7$ before and after $CO_2RR$ at the potential of −1.40 V, indicating the coexistence of CuPO and metallic Cu components in both $Cu_3(PO_4)_2$ and $Cu_2P_2O_7$ samples during $CO_2RR$. **c** HRTEM images of the $Cu_3(PO_4)_2$ sample after $CO_2RR$ at the potential of −1.40 V. **d** Enlarged HAADF-STEM image of $Cu_3(PO_4)_2$ after reaction and **e–g** its corresponding EDS elemental mapping images of **e** Cu, **f** P and **g** mixed elements, respectively. **h** HRTEM images of the $Cu_2P_2O_7$ sample after $CO_2RR$ at the potential of −1.40 V. **i** Enlarged HAADF-STEM image of $Cu_2P_2O_7$ after reaction and **i–l** its corresponding EDS elemental maps of **j** Cu, **k** P and **l** mixed elements, respectively, displaying the nanometric Cu/CuPO architecture in both $Cu_3(PO_4)_2$ and $Cu_2P_2O_7$.

d-spacing of (210) and (012) plane (Fig. 2c). False color further highlights the close contact between metallic Cu and $Cu_3(PO_4)_2$ (Supplementary Fig. 13a). As shown in the enlarged HAADF- STEM image (Fig. 2d) and the element mappings (Fig. 2e–g), the segregated Cu nanoparticles are observed, which is in agreement with the XRD result. As for $Cu_2P_2O_7$, the TEM images show that metallic Cu nanoparticles with sizes ranging from 3 - 25 nm that are featured by the d-spacing of (111) planes are distributed on $Cu_2P_2O_7$ that are featured by the d-spacing of (022) planes after the $CO_2RR$ process (Fig. 2h and Supplementary Fig. 12b). The distribution of metallic Cu and $Cu_2P_2O_7$ was further confirmed by the false-color HRTEM (Supplementary Fig. 13b) and HAADF-STEM images (Fig. 2i) overlapped with the corresponding EDS element mappings (Fig. 2j–l). By contrast, the TEM images of the CuO control sample after $CO_2RR$ show that the pristine CuO phase has been reduced to $Cu^0$ during the test (Supplementary Fig. 14).

In order to identify the variation of surface valence states and chemical compositions of CuPO and CuO samples after $CO_2RR$, X-ray photoelectron spectroscopy (XPS) was conducted (Supplementary Figs. 15–18). As observed in Supplementary Fig. 15, the peaks detected at about 933.2 and 935.5 eV of initial $Cu_2P_2O_7$ and CuO can be ascribed to the $Cu^{2+}$ component[33]. After 1 h's $CO_2RR$ at −1.40 V, the characteristic $Cu^0/Cu^{1+}$ (932.5 eV) peak appears and occupies a dominant position in CuO, indicating the surface $Cu^{2+}$ components of the CuO sample are mainly reduced to $Cu^0/Cu^{1+}$ (ref. 33). While the $Cu_2P_2O_7$ sample continues to exhibit the main component of $Cu^{2+}$ species after $CO_2RR$, which benefits from the stability at a negative potential of $Cu_2P_2O_7$. Due to the fact that the binding energies of the $Cu^0/Cu^{1+}$ states are difficult to distinguish in the Cu 2p region, we further performed a Cu LMM auger peak analysis (Supplementary Fig. 16). The peaks at around 567.7 and 570.0 eV demonstrate that both $Cu^0$ and $Cu^{1+}$ are formed in $Cu_2P_2O_7$ and CuO after $CO_2RR$[34]. Furthermore, the feature around 570.4 eV confirms the dominant existence of $Cu(OH)_2$ (ref. 34), indicating the component $Cu^{2+}$ species are preserved in the $Cu_2P_2O_7$ catalyst after $CO_2RR$. For the spectrum in the P 2p region (Supplementary

Fig. 17a), the characteristic P $2p_{1/2}$ and $2p_{3/2}$ peaks for $P_2O_7^{4-}$ are detected at the BE of 134.8 and 133.8 eV, indicating the phosphate group persists during the test[35]. Further evidence is derived from the O 1s spectrum, the peak located at 531.3 eV can be assigned to the non-bridging O in the phosphate group (P=O), and the peak at the BE of 533.1 eV can be attributed to the symmetric bridging O in P−O−P group (Supplementary Fig. 17b)[35]. Fourier-transformed infrared (FTIR) and Raman spectroscopy are two powerful methods to analyze low-frequency modes in the phosphates system. As shown in the FTIR spectrum of $Cu_2P_2O_7$, characteristic peaks that appear in 1100–900 cm$^{-1}$ region are attributed to symmetric and asymmetric P–O bonds, as well as the asymmetric P–O–P bridge vibration, confirming the phosphate group in the sample before and after $CO_2RR$ (Supplementary Fig. 19)[36]. The Raman spectrum of $Cu_2P_2O_7$ is displayed in Supplementary Fig. 20a, peaks located at the frequency areas of 1250–950 cm$^{-1}$ are known to be assigned to the P–O stretching modes of $[P_2O_7]^{4-}$. The symmetric and asymmetric stretch of P–O–P bridge appear in 930–970 and 680–760 cm$^{-1}$ regions respectively, and characteristic peaks observed in the area of 600–500 cm$^{-1}$ ($PO_2^{2-}$ radical) and 500–370 cm$^{-1}$ (P–O–P bridge) correspond to the P–O–P bending vibration[37]. In addition, the P–O–P deformations, the $PO_3$ rocking and deformation modes, and the external and torsional modes are detected in the region of 430–180 cm$^{-1}$ (ref. 38). These characteristic peaks are confirmed as pyrophosphate compounds, and they further demonstrate the durability of the pyrophosphate group during $CO_2RR$. For comparison, the Raman spectra of CuO are given in Supplementary Fig. 20b, and the bands found at 293, 345, and 633 cm$^{-1}$ are attributed to the standard Ag, Bg (1), and Bg (2) mode, respectively[39]. After being applied at −1.40 V for 1 h, no obvious peak could be detected in the Raman spectra, which proves that the CuO was totally reduced to Cu(0) during $CO_2RR$. We then implemented in situ Raman spectroscopy on the $Cu_3(PO_4)_2$ catalysts, which were measured under a constant potential of −1.40 V in the $CO_2$-saturated 0.1 M $KHCO_3$ electrolyte. Figure 3a shows the time-dependent in situ Raman spectra

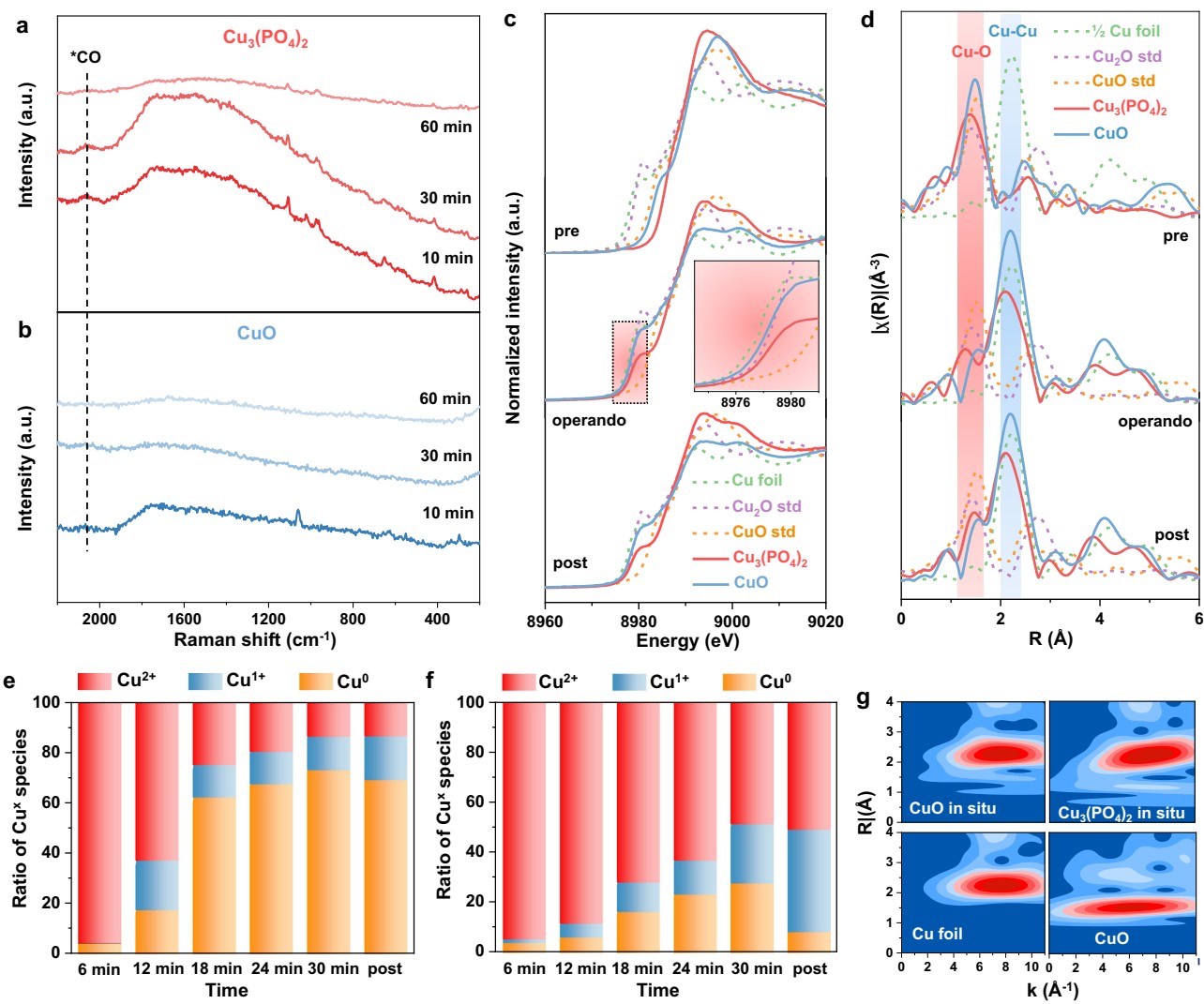

**Fig. 3 | Structural evolution investigation of CuPO and CuO catalysts during CO₂RR.** In situ Raman spectra of **a** Cu₃(PO₄)₂ and **b** CuO catalysts, respectively, during CO₂RR at −1.40 V in CO₂-saturated 0.1 M KHCO₃, suggesting the existence of Cu₃(PO₄)₂ and the disappear of CuO during CO₂RR. XAFS characterization of **c** the normalized Cu $K$-edge operando XANES spectra, **d** Fourier-transformed Cu $K$-edge EXAFS spectra, and **e, f** Calculated ratio of Cu oxidation states in **e** CuO and **f** Cu₃(PO₄)₂ catalysts from linear combination fitting with respect to time during 30 min of reaction at −1.45 V. The in situ Raman spectra along with operando XAFS analysis demonstrate the coexistence of Cu²⁺ and Cu⁰ components in Cu₃(PO₄)₂ and the reductive process of oxide Cu species to the metallic Cu⁰ states in the CuO control sample. **g** Morlet WT of the $k^3$-weighted operando EXAFS data for the Cu₃(PO₄)₂ and CuO samples with standard Cu foil and CuO powder as controls, suggesting the coexistence of Cu–O and Cu–Cu bond in Cu₃(PO₄)₂ during CO₂RR.

of Cu₃(PO₄)₂. The stretching of the PO₄³⁻ unit is observed at around 1000–1100 cm⁻¹, and the bands around 450–650 cm⁻¹ are attributed to the bending vibration of the PO₄³⁻ unit[38]. Considering that a moderate decrease in Raman spectra is observed in 60 minutes (min), we concluded that metallic Cu⁰ formed as the reduction of Cu₃(PO₄)₂ started, whereas residual Cu₃(PO₄)₂ remained present in the bulk. Noticeably, the atop-adsorbed CO (CO_atop) peak, acting as an important intermediate for the C–C coupling process, was detected at about 2060 cm⁻¹ (ref. [40]). As for the CuO control sample in Fig. 3b, the Raman peaks of CuO disappeared after 30 min, which indicated its entire reduction; moreover, no obvious *CO peak was detected. The in situ Raman measurement result suggests that sufficient coverage of CO* could be built on the Cu/Cu₃(PO₄)₂ catalyst, promoting further dimerization during CO₂RR.

To explore the oxidation state evolution of the electrocatalysts during CO₂RR, we carried out operando XAFS measurements under the electrochemical condition at −1.45 V (details in the Methods, Supplementary Fig. 21). The operando X-ray absorption near edge structure (XANES) spectra clearly show that the CuO sample appears

metallic Cu⁰ during CO₂RR and kept its metallic Cu⁰ state after test, while Cu₃(PO₄)₂ stays in higher valence states than metallic Cu(0) in the whole course of CO₂RR (Fig. 3c). Figure 3d exhibits the operando extended region of the XAS (EXAFS) spectra, and peaks at about 1.5 Å and 2.1 Å are attributed to the coordination of Cu–O and Cu–Cu, respectively[41,42]. A distinct peak of Cu–O coordination can be detected in CuPO at about 1.4 Å, indicating the existence of oxygen-bearing Cu species, and the peak shifts to a smaller radial distance gradually when CO₂RR takes place, implying the partial reduction of Cu²⁺. The enhancement of peak (Cu–Cu) in the samples (solid lines) suggests the emergence of Cu⁰, and Cu₃(PO₄)₂ catalyst shows a higher Cu–O coordination number and a lower Cu–Cu coordination number compared with CuO indicating the stability of Cu₃(PO₄)₂ at a negative potential to a certain extent. Meanwhile, we fitted the EXAFS data of Cu₃(PO₄)₂ during CO₂RR, and the fitting results further show that Cu–O coordination and Cu–Cu coordination exist simultaneously in the Cu₃(PO₄)₂ catalyst under reaction conditions (Supplementary Fig. 22 and Table 1). To quantitatively analyze the evolution of oxidation states during CO₂RR, we processed in situ spectra using an LCF of

the CuO and $Cu_3(PO_4)_2$ samples (Fig. 3e, f). The LCF spectra and data prove that the fitting results are reliable (Supplementary Fig. 23 and Table 2). From the LCF results, we calculated the ratios of Cu oxidation species presented at each 6 min under −1.45 V. After 6 min, the majority of CuO (96%) and $Cu_3(PO_4)_2$ (95%) were both in the $Cu^{2+}$ oxidation state, while they had respectively decreased to 25% and 72% after a further 12 min, which revealed that the transition between $Cu^{2+}$ and $Cu^0$ for CuO was more rapid, while the reduction for $Cu_3(PO_4)_2$ was slower. After the catalysts had been electrochemically reduced via $CO_2RR$ for 30 min, the majority of the CuO sample had been almost entirely reduced to $Cu^0$ (73%), while the $Cu_3(PO_4)_2$ sample was still 49% composed of the $Cu^{2+}$. These results show that $Cu^{2+}$ of $Cu_3(PO_4)_2$ may be stabilized so that $Cu^{2+}$ and $Cu^0$ could be coexistent under −1.45 V for over 30 min. Solid support for the coexistent metallic $Cu^0$ and $Cu^{2+}$ during the $CO_2RR$ process in $Cu_3(PO_4)_2$ was also provided through the additional analysis of the Morlet wavelet transform (WT). As shown in Fig. 3g, a WT maximum at $6-8\,\text{Å}^{-1}$ assigned to Cu–Cu bond is visible in the CuO sample during $CO_2RR$ at −1.45 V, which indicates the metallic $Cu^0$ species occupy the main composition in the bulk during the test. By contrast, the $Cu_3(PO_4)_2$ catalyst exhibits a distinct feature at $5-10\,\text{Å}^{-1}$ besides the WT maximum at $6-8\,\text{Å}^{-1}$, which represents the coexistence of Cu–O and Cu–Cu bond, demonstrating the component of Cu oxidation species still remained under the same reaction condition[41]. The operando XAFS analysis in conjunction with in situ Raman results clearly proved the specific coexistence of $Cu^{2+}$ and $Cu^0$ species in CuPO samples under $CO_2RR$ condition which could affect the selectivity.

### Electrochemical $CO_2$ conversion

The activity of CuPO catalysts for electrocatalytic $CO_2RR$ was evaluated in an H-cell with $CO_2$-saturated 0.1 M $KHCO_3$ solution as an electrolyte, and the electrodes were prepared by dropping casting ink solution onto the glassy carbon electrodes (GCE, see more details in the "Methods"). The linear sweep voltammetry (LSV) curves of the $Cu_3(PO_4)_2$/GC, $Cu_2P_2O_7$/GC, and CuO/GC in the $CO_2$-saturated (full line) or Ar-saturated (dotted line) environments are shown in Fig. 4a. The $Cu_3(PO_4)_2$ and $Cu_2P_2O_7$ catalysts exhibited larger geometric reduction $j$ in the $CO_2$ atmosphere, which demonstrated the reaction priority to $CO_2RR$. Thereafter, $Cu_3(PO_4)_2$/GC, $Cu_2P_2O_7$/GC, and CuO/GC were measured over a range of potentials for catalytic activity (Supplementary Fig. 24). $Cu_3(PO_4)_2$/GC shows a marked selectivity for $C_2H_4$ ($FE_{C2H4} > 60\%$) under potentials from −1.25 to −1.50 V, with a topmost $FE_{C2H4}$ reaching 69.7% at −1.45 V (Fig. 4b), corresponding to a $j_{C2H4}$ of −23.0 mA cm$^{-2}$, similar to what is tested on $Cu_2P_2O_7$/GC ($FE_{C2H4}$ of 64.0% and $j_{C2H4}$ of −17.6 mA cm$^{-2}$ at −1.40 V (Fig. 4c). As a comparison, the CuO/GC is less selective and its $CO_2RR$ catalysis yields CO (<8% FE) and $CH_4$ (<18% FE), as well as $C_2H_4$ (<8% FE) in −1.20 ~ −1.50 V (Fig. 4c and Supplementary Fig. 24). In particular, the ratios of $C_2H_4$ and $C_1$ products for $Cu_3(PO_4)_2$/GC and $Cu_2P_2O_7$/GC reach about 12.4 and 7.8 at −1.5 V, which is overwhelmingly superior to CuO/GC (Fig. 4d). Specific values for all products are given in Supplementary Tables 3 and 4.

To further improve the $CO_2RR$ current density, we constructed gas diffusion electrodes (GDE) with the $Cu_3(PO_4)_2$ catalyst deposited on the Cu-coated polytetrafluoroethylene gas diffusion layers ($Cu_3(PO_4)_2$/Cu/PTFE) and evaluated $CO_2RR$ performance utilizing a flow cell (details in the "Methods"). The 2.0 M KOH alkaline electrolyte was used to enhance the conductivity and improve the $CO_2RR$ kinetics by suppressing $H_2$ evolution. As shown in Fig. 4e, the $Cu_3(PO_4)_2$/Cu/PTFE catalyst was screened at different current densities and reached a maximum $FE_{C2+}$ of 90.9% (52.8% ethylene, 30.7% ethanol, 5.9% acetic acid, and 1.5% n-propyl alcohol) with $j_{c2+}$ of −318.2 mA cm$^{-2}$ corresponding to experiments performed at $j_{tot} = -350$ mA cm$^{-2}$. Supplementary Fig. 25 exhibits the chronopotential curves obtained at different $j$. Meanwhile, the bare Cu/

PTFE showed $FE_{C2+}$ of 58.9 ~ 77.2% with the $j_{tot}$ ranging from −100 to −400 mA cm$^{-2}$ (Supplementary Fig. 26), proving the $Cu_3(PO_4)_2$ catalyst has excellent $CO_2$-to-$C_{2+}$ conversion performance intrinsically. Electrochemical stability is also a key parameter for $CO_2RR$ performance. Remarkably, $Cu_3(PO_4)_2$/Cu/PTFE unveils a desirable stability in the flow cell during $CO_2RR$ for 18 h (Fig. 4f). The total current density keeps at −300 mA cm$^{-2}$, and $FE_{C2+}$ remains above 80% after 18 h of reaction, representing that the catalyst is stable during $CO_2RR$. Furthermore, we compared the maximum $FE_{C2+}$ of 90.9% for $Cu_3(PO_4)_2$/Cu/PTFE with that for other Cu-based electrocatalysts (Fig. 4g and Supplementary Table 5), which was found to possess a remarkable $C_{2+}$ production selectivity compared to the reported Cu-based catalysts[10,12,17,43-57].

### Insights into $CO_2$-to-$C_{2+}$ electroreduction

The $CO_2RR$ intermediates chemisorbed on CuPO and CuO were assessed via SEIRAS to determine the mechanism for boosted $C_{2+}$ selectivity (details in the Methods, Supplementary Fig. 27). As shown in Fig. 5a, the in situ SEIRAS differential spectra of $Cu_3(PO_4)_2$ exhibit peaks at about 2065 cm$^{-1}$ and 1025 cm$^{-1}$, which are associated with the absorbed *CO and the nonplanar vibration (O=C–H) of *CHO on the catalyst surface, respectively[58,59]. Additionally, the peaks detected at about 1255 and 1385–1410 cm$^{-1}$ could be attributed to the C–O stretch and symmetric vibration (vibration of O–C=O) of *COOH, respectively[59]. Prominently, based on reported experimental and theoretical studies, the distinctive 1180 and 1520 cm$^{-1}$ peaks can demonstrate the existence of *OCCHO intermediate; it increased on scanning to more negative potentials, which is consistent with the trend of $C_{2+}$ products' formation rates[59-61]. To support the band assignments of the IR to *OCCHO theoretically, we simulated the IR of *OCCHO intermediates that adsorbed at the $Cu^0/Cu^{2+}$ ($Cu_3(PO_4)_2$) interface. As shown in Supplementary Fig. 28, a band peak with strong oscillator strength is located at 1189 cm$^{-1}$, which is close to our experiment result (1180 cm$^{-1}$) for *OCCHO. Moreover, the peak intensity of *CO peaks on $Cu_3(PO_4)_2$ enhances first as *CO increases, and then gradually decreases as the applied potential continues to turn negative, suggesting the *CO is consumed as dimerization accelerates. In comparison, the absorbed *CO continues to accumulate and the C–C coupling bond is not detected on the CuO sample, which would be entirely reduced to $Cu^0$ during $CO_2$ reduction (Fig. 5b). As illustrated in the schematic diagram (Fig. 5c), the in situ SEIRAS study in conjunction with theoretical research reveals that the stable and abundant $Cu^0/Cu^{2+}$ interfaces derived from Cu phosphate-based electrocatalysts can facilitate the pathway of *CHO coupling with *CO to form *OCCHO, thus improving the selectivities of $C_{2+}$ products during $CO_2$ reduction.

## Discussion

In summary, we have theoretically predicted and experimentally constructed the in situ formation of nanometric Cu/CuPO with rich and stable $Cu^{0+}/Cu^{2+}$ interfaces to be the efficient electrocatalyst for highly selective $CO_2$-to-$C_{2+}$ products conversion. The resultant nanometric Cu/CuPO could obtain a high $FE_{C2H4}$ of 69.7% in a neutral medium and perform a maximum $FE_{C2+}$ above 90% with industrially relevant current densities in a flow cell configuration. The $Cu^{2+}$ sites of $Cu^{0+}/Cu^{2+}$ interfaces were revealed to be available for the formation of *CHO, which then facilely coupled with *CO on the adjacent $Cu^0$ surface to form *OCCHO, leading to high-efficiency $CO_2$-to-$C_{2+}$ conversion performance. Our work highlights the role of the peculiar $Cu^0/Cu^{2+}$ interfaces in promoting selectivity toward $C_{2+}$ products, and we believe that this finding will contribute to the design of improved Cu-based electrocatalysts for future $CO_2RR$, with high selectivity and underlying mechanisms for its conversion to valuable products.

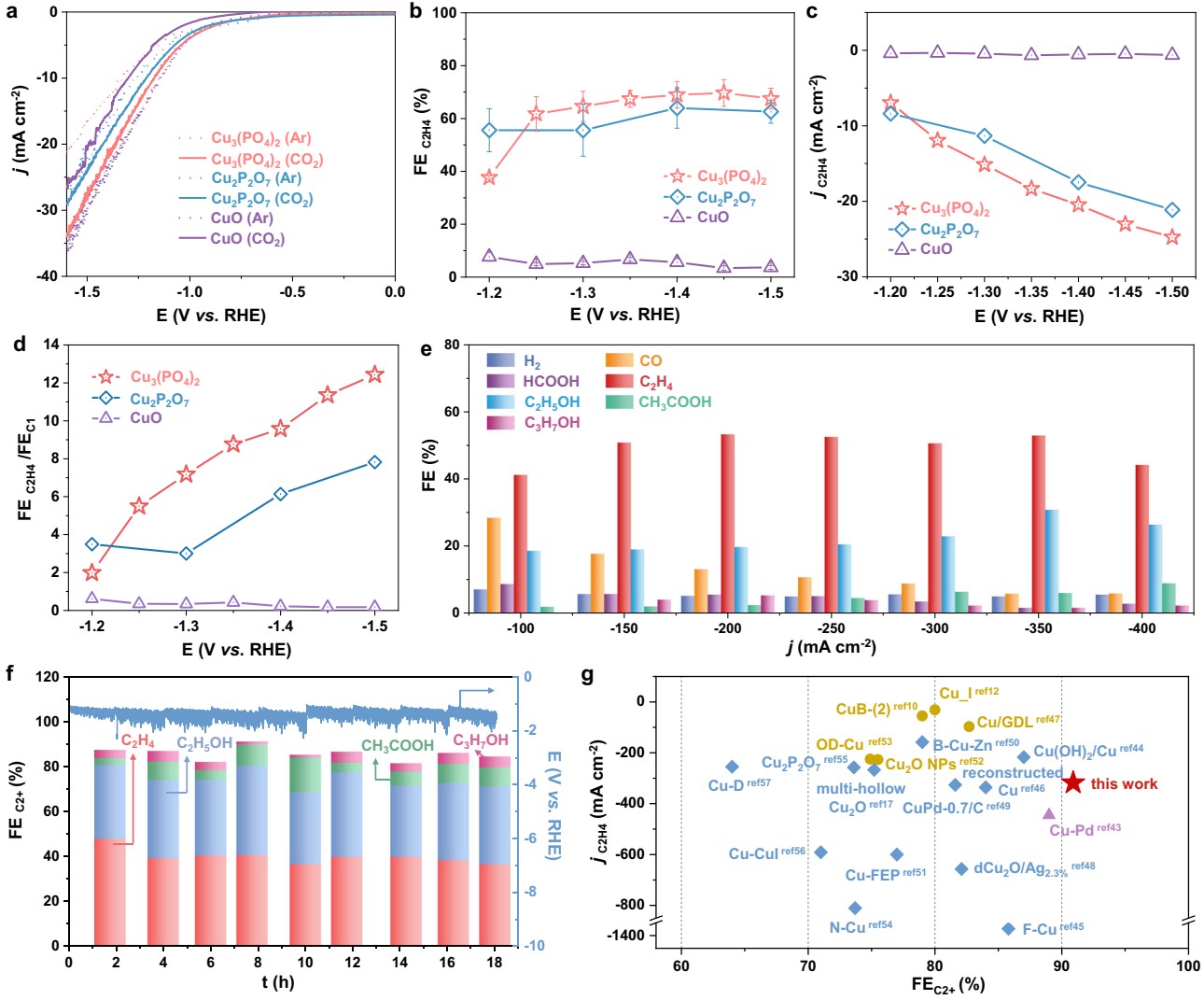

**Fig. 4 | Electrochemical performance evaluation of CuPO and CuO catalysts for CO₂RR. a** LSV curves of CuPO/GC and CuO/GC catalysts with respect to RHE at a scan rate of 1 mV/s in the CO₂- or Ar-saturated 0.1 M KHCO₃ electrolyte, indicating the occurrence of CO₂RR. **b** FEs of C₂H₄ of catalysts at different applied potentials in CO₂-saturated 0.1 M KHCO₃, suggesting the superior C₂H₄ selectivity for CuPO. Error bars above were all based on the standard deviation of three measurements at each potential. **c** Potential dependence of $j_{C2H4}$ on CuPO/GC and CuO/GC, exhibiting higher C₂H₄ activity for CuPO. **d** FE$_{C2H4}$/FE$_{C1}$ ratios of catalysts at different applied potentials, suggesting the superior C₂H₄ selectivity for CuPO/GC.

**e** FEs for products over Cu₃(PO₄)₂/Cu/PTFE at various applied current densities in a flow cell reactor with 2.0 M KOH as the electrolyte. **f** FEs of various products produced by Cu₃(PO₄)₂/Cu/PTFE at the applied $j$ of −300 mA cm⁻² for 18 h in the 2.0 M KOH electrolyte (refresh electrolyte for every 2 h), and the liquid products were collected after the reaction run for more than 30 min. **g** Maximum FE$_{C2+}$ of Cu₃(PO₄)₂/Cu/PTFE and recently reported Cu-based CO₂RR catalysts (details in Supplementary Table 5). Blue rhombus: alkaline medium; yellow circle: neutral medium; violet triangle: acidic medium.

## Methods

### DFT calculation

All the calculations were performed using the Vienna Ab-initio Simulation Package (VASP) package[62–64]. The exchange-correlation functional was described by the Perdew–Burke–Ernzerhof (PBE) functional[65] within the generalized gradient approximation (GGA)[66]. The project-augmented wave (PAW) method[67] was employed to treat core electrons, and the cut-off energy of the plane-wave basis was set to 450 eV. A vacuum layer of 15 Å was applied to separate each periodic unit cell. In the structural optimizations, the Brillouin zone was sampled by 2 × 2 × 1 Monkhorst–Pack mesh k-points, and the bottom two layers of the slab were fixed, and the top three layers and adsorbates were fully relaxed. The empirical correction in Grimme's scheme was used to describe the van der Waals interactions[68]. Considering the solvation environment, we constructed an explicit solvation model with one layer H₂O molecules in calculation models.

### Catalyst preparation

For Cu₃(PO₄)₂, 12 mmol Cu(NO₃)₂·3H₂O and 8 mmol NH₄H₂PO₄ were dissolved in H₂O and stirred well to form a suspension in a beaker. Then, added 6 mmol C₆H₁₀O₈ to beaker and stirred until the suspension was clear. The beaker was completely covered with tin foil, with some small holes punched in the top, then placed in the oven and dried at 120 °C. Finally, the dried material was transferred to a crucible and calcined at 700 °C for 1 h in a muffle furnace at a heating rate of 5 °C/min. The preparation method of Cu₂P₂O₇ was the same as the above procedure except for changing the mass of Cu(NO₃)₂·3H₂O to 8.5 mmol. The preparation method of CuO was the same as that of Cu₃(PO₄)₂ except that NH₄H₂PO₄ was not added. All reagents were commercially available as analytical grade (Supplementary Note 3).

### Working electrodes preparation

For the Cu₃(PO₄)₂/CP, Cu₂P₂O₇/CP, and CuO/CP used for XRD, XPS, Raman, FTIR, and XAFS measurements. The sample ink was prepared

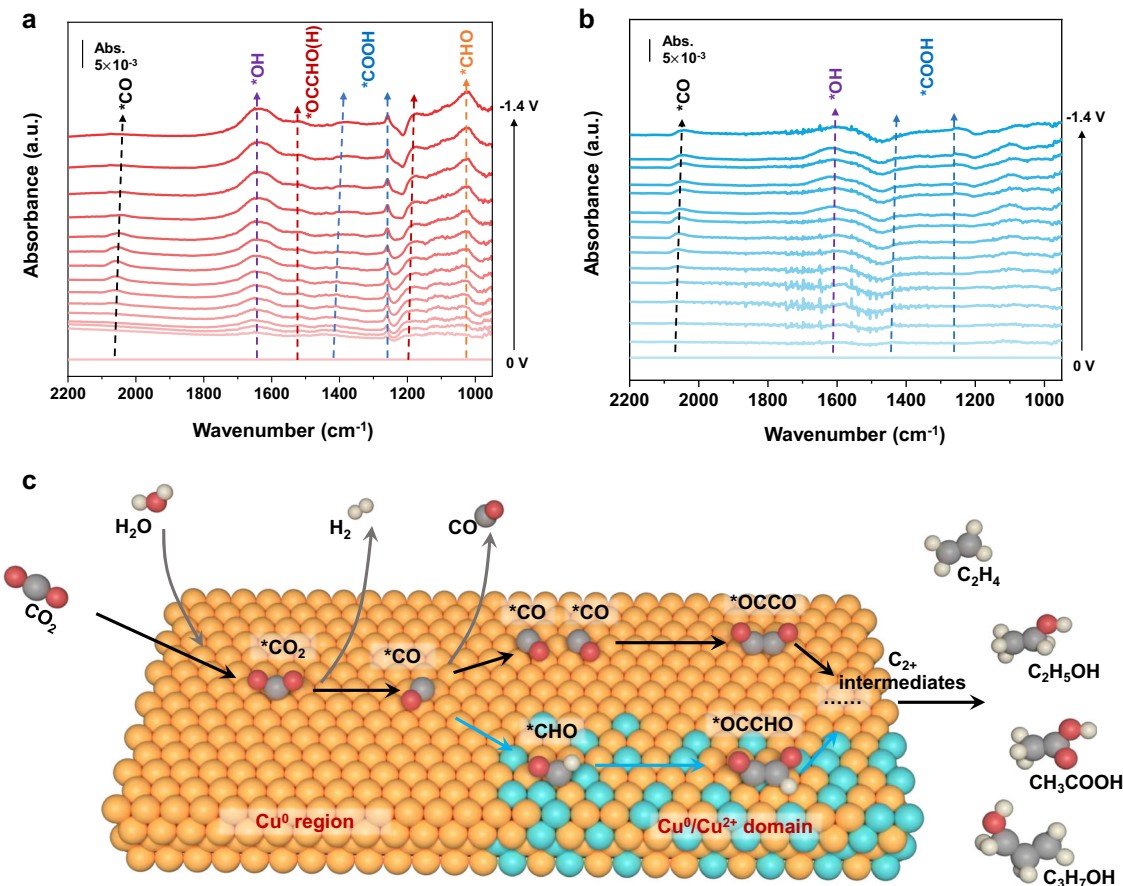

**Fig. 5 | Insights into CO₂-to-C₂₊ electroreduction.** In situ SEIRAS differential spectra of the **a** Cu₃(PO₄)₂ and **b** CuO sample in CO₂ purged 0.1 M KHCO₃ electrolyte in real-time condition. **c** Schematic illustration on proposed role of Cu(II)/ Cu(0) interfaces in CO₂-to-C₂₊ conversion. Red: oxygen; gray: carbon; white: hydrogen; origin: Cu⁰; golden: Cu²⁺.

by dispersing 10 mg of catalyst ($Cu_3(PO_4)_2$, $Cu_2P_2O_7$, or CuO) and 100 µL of Nafion solution into 0.5 mL of $H_2O$ and 1.5 mL of iso-propanol, followed by sonication for more than 1 h. The sample ink was sprayed on carbon paper using an air-brush, and the total catalyst loading was about 2.0 mg cm$^{-2}$. For the GCE working electrode used in H-Cell electrochemical measurements, the sample ink was prepared by dispersing 5 mg catalyst ($Cu_3(PO_4)_2$, $Cu_2P_2O_7$ or CuO) and 50 µL Nafion solution into 0.125 mL $H_2O$ and 0.375 mL iso-propanol followed by sonication for more than 30 min. 2.5 µL ink was dropped on GCE (with a diameter of 3 mm) and dried in the air for 2 times, with the total loading amount of catalyst about 0.7 mg cm$^{-2}$.

For Cu/PTFE, bare PTFE with an average pore diameter of 0.22 µm was used. Approximately 600 nm nominal thick Cu films were constructed by vacuum evaporation method on the PTFE substrate using Cu target material (99.999%) at an evaporation rate of around 0.5 Å s$^{-1}$ in an OMV FS300-S6 evaporating tool at a base pressure of <6*10$^{-4}$ Torr. For $Cu_3(PO_4)_2$/Cu/PTFE, the sample ink was prepared as above mentioned on carbon paper and sprayed on the Cu/PTFE films electrodes using air-brush, with the total loading amount of catalyst about 1.0 mg cm$^{-2}$.

**Materials characterizations**

The crystal structure was determined using X-ray diffraction (Bruker D8 Advanced Diffractometer with Cu Kα radiation). The morphology and structure were characterized by scanning electron microscope (Hitachi S4800) and transmission electron microscopy (TEM, JEOL JEM 2010, operated at 200 kV). Scanning transmission electron microscopy (STEM) characterization was performed using a ThermoFisher

Talos F200X. High-angle annular dark-field (HAADF)-STEM images were recorded using a convergence semi angle of 11 mrad, and inner- and outer collection angles of 59 and 200 mrad, respectively. Energy-dispersive X-ray spectroscopy (EDS) was carried out using 4 in-column Super-X detectors. The chemical state was analyzed by X-ray photo-electron spectroscopy (XPS, Thermo Escalab 250), and the binding energy of C 1s peak at 284.8 eV was taken as an internal standard. Raman analysis was carried out using a Leica DMLM microscope (Renishaw) with the 514 nm laser. FTIR spectroscopy was characterized on a Nicolet 6700 spectrometer with a spectral range of 4000–400 cm$^{-1}$. XAFS spectra at the Cu K-edge were performed on the 1W1B beamline station of the Beijing Synchrotron Radiation Facility (BSRF), China. Cu foil, $Cu_2O$ and CuO were used as references. LCF were processed using the ATHENA module implemented in the IFEFFIT software packages.

The in situ XAS measurement was performed in a homemade plastic electrolytic cell. The graphite rod (spectral purity, 3 mm in diameter) and the Ag/AgCl electrode acted as the counter electrode and the reference electrode in the three-electrode system, respectively. The carbon paper coated with specific catalysts was used for the working electrode, and a gas inlet for purging $CO_2$ into the electrolyte (0.1 M KHCO₃) was also contained in the cell. A window (1.1 cm × 1.8 cm) was designed on the cell for the working electrode that enabled XAS measurement under a sensitive fluorescence model while the working potentials were applied. A spectrum can be acquired on average every 1.5 min, and the in situ curves of the samples in Fig. 3c, d were taken at around 36 min of reaction.

The in situ Raman measurement was performed on the Raman spectrometer (LabRAM HR) utilizing an excitation laser with a wavelength of 514 nm and a 50× microscope objective with a numerical aperture of 0.5, 10.6 mm. Before the experiments, calibration was carried out based on the peak at 520 cm$^{-1}$ of a silicon wafer standard. To acquire information about the electroreduction process of Cu$_3$(PO)$_4$ and CuO, we utilize a spectroelectrochemical cell and detect the in situ Raman of the cathode GDE through a quartz window. The GDEs of Cu$_3$(PO)$_4$ and CuO were taken as the working electrodes, which were prepared by the catalyst sprayed on carbon paper. Platinum wire and Ag/AgCl were used as the counter electrode and reference electrode, respectively. During the in situ experiment, a peristaltic pump was used to control the flow rate of CO$_2$-saturated 0.1 M KHCO$_3$ electrolyte at 10 mL/min, while the flow rate of CO$_2$ was kept at 10 sccm with a mass flow controller.

In situ surface-enhanced infrared absorption spectroscopy (SEIRAS) was recorded in a homemade reflection accessory with internal reflection configuration using an FTIR spectrometer equipped with a PerkinElmer spectrum 100 detector. FTIR spectra were obtained by averaging 16 scans with a resolution of 8 cm$^{-1}$ at the selected potential. Every spectrum was obtained by applying a single potential step compared to the reference potential (0 V). For the spectroelectrochemical measurements, a thin Au film with a thickness of ~10 nm is prepared on a Si prism using electroless deposition[69]. The sample ink was prepared as mentioned above on GCE. 25 μL ink was dropped on Au underlayer and dried in the air for 2 times. The catalyst-coated Si crystal was placed in a three-electrode spectroelectrochemical cell as the working electrode, and the counter and reference electrodes were platinum wire and Ag/AgCl electrode (3.5 M KCl), respectively. A gas inlet for purging CO$_2$ was also contained in the cell. Electrolyte (0.1 M KHCO$_3$) was injected into the cell and gassed with high-purity CO$_2$ (99.9999%) prior to electrochemical measurements. A CHI1242C potentiostat was used to record electrochemical responses. Spectra were expressed as absorbance, with positive and negative peaks showing an increase and decrease in signal, respectively.

## Electrochemical measurements

All electrochemical studies were performed using an electrochemical station (CHI 660E). The H-type gas-tight reactor consisted of two compartments separated by a Nafion 117 membrane. In the three-electrode system, a custom-made GCE was used as working electrode with a surface area of 0.07 cm$^2$ and a catalyst mass loading of 0.7 mg cm$^{-2}$. Ag/AgCl electrode (3.5 M KCl) and platinum mesh were used as reference electrode and counter electrode, respectively. CO$_2$-saturated 0.1 M KHCO$_3$ (pH ≈ 6.75) was used as electrolyte. All measured potentials were calibrated to the RHE reference scale using $E_{RHE} = E_{Ag/AgCl} + 0.059 \times pH + 0.205$ (all potentials were not iR-corrected if not mentioned). Linear sweep voltammetry (LSV) at a scan rate of 1 mV s$^{-1}$ were performed in Ar and CO$_2$-saturated 0.1 M aqueous KHCO$_3$. The current density was calculated by normalizing the current to the corresponding geometric surface area.

The electrocatalytic performance of Cu$_3$(PO$_4$)$_2$ was determined in a flow cell configuration. The device consists of Cu$_3$(PO$_4$)$_2$/Cu/PTFE as the working electrode, nickel foam as the anode and an anion exchange membrane. They were assembled using PTFE gaskets with 2.0 M KOH as the liquid electrolyte flowing (10 mL per min) in the chambers between membrane and working electrode, and membrane and anode. CO$_2$ gas was flowed behind the PTFE layer at a rate of 20 sccm, which was decreased to 19 sccm at the outlet due to the consumption of the alkaline electrolyte. Chronopotentiometry experiments were performed at currents of −100 mA cm$^{-2}$, −150 mA cm$^{-2}$, −200 mA cm$^{-2}$, −250 mA cm$^{-2}$, −300 mA cm$^{-2}$, −350 mA cm$^{-2}$, and −400 mA cm$^{-2}$,

showing consistent gas product distributions. The liquid products were collected after the reaction run for more than 30 min.

## Products analysis

The gas products of CO$_2$ electroreduction were analyzed by gas chromatography (GC online test, RAMIN, GC2060), equipped with a flame ionization detector (FID to detect CO, CH$_4$ and C$_2$H$_4$) and a thermal conductivity detector (TCD to detect H$_2$). Ar was used as a carrier. CO$_2$ was continuously sparged through the electrolyte (30 mL of catholyte and anolyte in each compartment) at a rate of 20 sccm and was routed into the gas chromatograph. The Faradic efficiencies were tested online and averaged for multiple data.

The liquid products were quantified by $^1$H nuclear magnetic resonance (NMR) (Varian 700 MHz spectrometer, 16.4 T). In a typical analysis, the mixture of 500 mg of the electrolyte and 100 mg of 149 ppm DMSO (used as internal standard) in D$_2$O solution was used as measured sample after 2 h of CO$_2$RR with $j$ applied at −300 mA cm$^{-2}$. The $^1$H spectra were obtained by suppressing the water peak using the pre-saturation method.

Assuming that 12 electrons are required to generate one C$_2$H$_4$ molecule, the FE and the partial current densities of C$_2$H$_4$ formation were calculated as below:

$$FE = \frac{12 F v G P_0}{i_{total} R T} \times 100\% \quad (1)$$

where $v$ = volume concentration of C$_2$H$_4$ obtained from gas chromatography (GC) data, $p_0 = 1.013$ bar and $T = 298.15$ K, $G = 20$ sccm is the CO$_2$ flow rate (19 sccm for 2 M KOH in a flow cell), $i_{total}$ (mA) = steady-state cell current, $F = 96485$ C mol$^{-1}$, $R = 8.314$ J mol$^{-1}$ K$^{-1}$. Then:

$$j_{C_2H_4} = FE_{C_2H_4} \times i_{total} \times (\text{electrode area})^{-1} \quad (2)$$

## Data availability

Source data are provided with this paper.

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

## Acknowledgements

This work was financially supported by the Key Program of National Natural Science Foundation of China (22239001), the International (Regional) Cooperation and Exchange Projects of the National Natural Science Foundation of China (51920105003), the National Natural Science Foundation of China (22379043, 22309053), the Science and Technology Commission of Shanghai Municipality (21DZ1207101, 22ZR1416400, 23YF1408500, 23ZR1416800), the Innovation Program of Shanghai Municipal Education Commission (E00014), the China Postdoctoral Science Foundation Funded Project (2022M721137), the Shanghai Engineering Research Center of Hierarchical Nanomaterials (18DZ2252400), and the Fundamental Research Funds for the Central Universities. The authors also thank the Frontiers Science Center for Materiobiology and Dynamic Chemistry. The authors also thank the crew of the BL14W1 beamline at the Shanghai Synchrotron Radiation Facility (SSRF) and the 1W1B beamline of Beijing Synchrotron Radiation Facility (BSRF) for their constructive assistance with the XAFS measurements and data analyses.

## Author contributions

H.G.Y. and P.F.L. directed the research. X.Y.Z. performed the experiments and data analyses. Z.X.L. performed the DFT calculations. M.Z. and J.C. assisted with the in situ SEIRAS tests and data analyses. S.D. and X.W. provided assistance with the HAADF-STEM characterization. Y.L. and J.Y.Z. performed the XAFS test and analyses. C.S., H.F.W. and H.Y.Y. guided the DFT calculations. All the authors participated in writing and editing the manuscript, and contributed their efforts to the discussion.

## Competing interests

The authors declare no competing interest.
