## [Peer Review File · Nature Communications]

REVIEWER COMMENTS

Reviewer #1 (Remarks to the Author):

The authors in this work describes a study in designing a Cu/Cu²⁺ catalyst for CO₂ electrolysis. They started by first computationally screening a list of candidates from the Materials Project database and identify Cu₂P₂O₇ and Cu₃(PO₄)₂ as promising candidates. The catalysts were synthesized via a sol gel method and electrochemically tested. The logic of the work is clear and well-written. The performance of the catalyst is very good. Although I do have some questions regarding the mechanism, I would support publication after suitable revision is provided.

- One of my major concern is the DFT calculation: majority of the DFT study is dedicated to comparing the energy profile for the CO-CO vs. COH-CO vs. CHO-CO pathways on the Cu(0) vs. Cu(I) vs. Cu(II) surface. However, all three of the reaction pathways are a dimerization step, which all lead to C₂+ products. And one of the major experimental claim is that the Cu(II) surface leads to high C₂+ yield, instead of C₁. The DFT study should've been conducted to compare the energy profile of C₁ vs. C₂+ reaction pathways on the Cu(0) vs. Cu(I) vs. Cu(II) surface.

- The authors used a P-doped synthetic strategy to stabilize a Cu(0)/Cu(II) surface. But there is not even a single mention of the catalytic effect of P during CO₂ electrolysis. I can't help but think – is the CO₂RR performance enhanced due to Cu(II) or the influence of the phosphorous element? It would be important for the authors to provide some experimental/computational data and discussion on this topic.

- All the characterization data (XRD, XPS, XAS) have detected some degree of reduction of the Cu(II) during electrolysis. While the reduction of Cu(II) is expected, the authors have provided data for 1 hour post reaction only. I am interested to know if the Cu(II) catalyst is stable for 5h, 10h or 20h? Can the author show a longer post reaction characterization to demonstrate if the Cu(II) would ever get stabilize or fully reduce?

- For the SEIRAS data, the authors assign the peaks at 1180 and 1520 cm⁻¹ to *OCCHO and *OCCHOH respectively, based on Ref 56 [Angew. Chem. Int. Ed. 2022, 61]. Ref 56 is a fully computational study that cites an experimental study [Angew. Chem. Int. Ed. 2017, 56, 3621]. In the experimental study that actually collected FTIR data, the peaks at ca. 1180 and 1520 cm⁻¹ were assigned to C–O–H and C=O stretching of OCCOH. To me, the second study is more reliable since it is actually an experimental study and it is what Ref 56 is based on. The difference in the peaks assignment do affect the conclusion to some degree. I hope the authors can cite the original study and revise the discussion accordingly.

- The authors used a Cu/PTFE as a substrate during the flow cell testing. Cu/PTFE itself is a very effective catalyst for C₂+ products during CO₂RR. The authors should provide the FE performance of Cu/PTFE alone as control.

Minor comments

- There are a number of oxide-derived Cu catalysts reported to-date (the authors have cited in the introduction). Also, there are P-doped [Appl. Surf. Sci. 2021, 544, 148965.], N-doped [Nat. Nanotechnol. 15, 131–137 (2020)], B-doped [Nature Chem 10, 974–980 (2018).] or cation-ion-doped [Nat Catal (2022) DOI:10.1038/s41929-022-00880-6] Cu catalysts that are somewhat similar to the catalysts reported in this study. So, I do find the novelty aspect of this work is lacking a bit. But the data in this work are complete and comprehensive, so this is not a major concern.

- Cu²⁺ should have a very signature satellite peak in XPS, that peak is almost completely missing in the CuO sample. Can the authors comments on the reason why?

- Line 350 and 351, the authors wrote "...high-efficiency CO₂-to-CO₂+ conversion...". I believe they meant "CO₂-to-C₂+ conversion..."

Reviewer #2 (Remarks to the Author):

Reviewer report for NCOMMS-22-48614

Direct OC-CHO coupling towards highly C₂+ products selective electroreduction over stable Cu₀/Cu²⁺ interface

Xin Yu Zhang^{1†}, Zhen Xin Lou^{1†}, Jiacheng Chen², Yuanwei Liu¹, Xuefeng Wu¹, Jia Yue Zhao¹, Hai Yang Yuan^{1*}, Minghui Zhu², Sheng Dai³, Hai Feng Wang⁴, Chenghua Sun⁵, Peng Fei Liu^{1*} & Hua Gui Yang^{1*}

This work fuels into the debate on Cu oxidation state during CO₂RR. The authors screened some Cu catalyst that may be able to stabilize Cu²⁺ and found a bunch of Cu-Phosphate catalyst to be used for electrochemical CO₂ reduction (CO₂RR) to multi carbon products. The authors attributed the high performance towards a stable Cu₀/Cu²⁺ interface, possibly due to the stability of Cu-Phosphate (either Cu₂P₂O₇ or Cu₃(PO)₄).

A key argument/novelty in this work is the detection of persistent Cu²⁺. Several supporting evidence is presented, namely XAS, in-situ Raman, and in-situ SEIRAS. Although interesting and the CO₂RR activity is decent, I think evidence backing the relevance of Cu²⁺ species in catalysis is not strong.

Detailed Feedback:

(1) Cu₂P₂O₇ use for CO₂RR has been reported before, (doi:10.1002/anie.202114238), and it does show tendency to produce multi carbon products more than oxide (either Cu₂O or CuO) derived Cu.

(2) It is confusing that the authors lumped both Cu₂P₂O₇ or Cu₃(PO)₄ as Cu-PO, yet the investigation of catalytic activity is focused more on Cu₃(PO)₄, and theoretical study uses Cu₂P₂O₇ Why is that?

(3) My main issue with this article is that the argument of persistent Cu²⁺ on Cu-PO catalysts, especially whether the said Cu²⁺ species is relevant catalytically is not convincing. Many literatures have used either Cu₂P₂O₇ or Cu₃(PO)₄ and state clearly that reconstruction can easily happen after cathodic potential application (see e.g. doi:10.1002/anie.202114238 doi:10.1016/j.electacta.2019.05.030). The authors must explain why they expect their Cu-PO catalysts can be stable and not reduced. I note the evidence included in this manuscript actually pointed clearly to major reduction to Cu:

a. XRD (Fig 2a, b) shows that, especially for Cu₃(PO)₄, the catalyst turn primarily to Cu.

b. EXAFS/XANES data shows significant reduction of the Cu₂P₂O₇ towards metallic Cu during (and possibly after) reduction. Key features in the near edge of Cu K edge (Fig 3c) shows clear pre-edge feature at lower energy and the oscillation start to resemble Cu₀ more than Cu¹⁺ or Cu²⁺.

c. The FT-XAS data in Fig 3d also shows disappearance of shorter Cu-O bonds, which indicates the destruction of Cu-O trigonal bipyramid in Cu₃(PO₄)₂ and Cu-Cu bonds start to dominate in the operando and post data.

d. Wavelet transform (Fig 3g) of Cu₃(PO₄) also shows more similar profile with CuO in situ, and Cu foil.

e. The authors rely very much on LCF to show that they have “Cu²⁺” residue, but unfortunately the fitting procedure, goodness of fit, and fitting compared to the spectra is not detailed. Where is the fit performed on? Is it in the energy space, k space or R space? It is actually hard to believe that, comparing the amount of Cu²⁺ shown in Figure 3f (approx. 50%), to Figure 3d mid graph (R space), that should actually show predominantly Cu₀ (marked by strong Cu-Cu). I believe it is more useful to do first (also second if the energy range is sufficient) shell fitting, to get the type of neighbouring atoms and approximate bond length of the first shell. It will be then more apparent whether this can be attributed to reduced Cu₀ or higher Cu oxidation (in form of oxides or phosphates).

f. Overall, only in-situ Raman shows existence of persistent PO₄ bands. Can this be a “spectator” effect, or unreduced catalyst at the edge of the sample (see e.g., 10.1021/ja960468v)? The authors may include more details on how the in-situ Raman is collected.

g. Figure 4a, shows reduction feature near 0 V vs RHE. This probably indicates Cu reduction. The authors should study the redox behaviour of both Cu₂P₂O₇ and Cu₃(PO)₄ to confirm if it is really stable under bias.

(4) It is interesting that *CO is observed in SEIRAS, but not visible in the in-situ Raman (considering IR are absorbed stronger in aqueous solution). I note that features of the in-situ SEIRAS is actually similar for

both $\text{Cu}_3(\text{PO})_4$ and CuO pre-catalysts. Similar signals are present, albeit weaker can be seen on CuO precatalyst, just not labelled.

(5) Raman figure x-axis were different for Fig 3a and 3b, making it difficult to compare between CuO and $\text{Cu}_3(\text{PO}_4)_2$ samples. It is also curious that $^*\text{CO}$ bands are absent even on CuO samples at cathodic potential (see e.g., doi:10.1021/acsami.8b08428, doi:10.1021/cs502128q)

(6) Theoretical calculation: the authors based their argument leaning towards Cu^{2+} sites, based on the ease of the protonation of $^*\text{CO}$ to $^*\text{CHO}$ or $^*\text{COH}$, on Cu , Cu_2O and CuO sites representing Cu^0 , $1+$, and $2+$ sites (Fig 1a). On higher Cu oxidation sites, the adsorption conformation of CO will be different due to presence of oxygen, and the first protonation step may not be the most difficult step. The authors should look at possibility of the initial CO_2 adsorption step difference between the Cu oxidation sites, and also with the Cu-PO surface.

a. Figure 1d: what is the meaning to the colour and circle size?

b. Reaction coordinate in Fig 1e-g should be compared to the classical $\text{Cu}(100)$ surface as baseline

(7) XPS (SI Fig 10-12), please expand the energy range visual to include the satellite, and include LMM scans, and probably construct a Wagner plot to distinguish Cu^0 , Cu^{1+} and Cu^{2+} species better

Reviewer #3 (Remarks to the Author):

This is a very interesting paper reporting a high-activity CuPO -based catalyst for electrochemical CO_2 reduction. The authors have made detailed comparisons with CuO and show convincing evidence that the new catalyst is of high activity. I have some comments, which should be considered before publication.

1. The band assignments of the IR to OCCHO etc. intermediates appear to be quite arbitrary. Do authors have some DFT frequency results to support this? Why they cannot be other similar C_2^+ species considering that so many different products are available? Some discussions are required.

2. It might be worth reporting the Tafel kinetics, which could help to confirm the rate-determining step, especially $\text{CO}+\text{CO}$, $\text{CO}+\text{CHO}$ are concerned (see review in *ACS Catal.*, 2014, 4, 4364).

3. It is not clear whether the solvation model is switched in DFT calculations, which is important for the solution reaction involving protons (see e.g. *J. Am. Chem. Soc.*, 2010, 132, 13008)

4. How about the long-term stability of Cu^{2+} in $(\text{Cu}_3(\text{PO}_4)_2)$? Could the stability be improved, i.e. without reducing to Cu^+ and Cu^0 ? The important role of PO_4 in protecting Cu^{2+} could also be modeled in DFT and clarified in the text.

Responses to Reviewers' Comments

Responses to Reviewer #1

General Comment:

The authors in this work describes a study in designing a Cu/Cu²⁺ catalyst for CO₂ electrolysis. They started by first computationally screening a list of candidates from the Materials Project database and identify Cu₂P₂O₇ and Cu₃(PO₄)₂ as promising candidates. The catalysts were synthesized via a sol gel method and electrochemically tested. The logic of the work is clear and well-written. The performance of the catalyst is very good. Although I do have some questions regarding the mechanism, I would support publication after suitable revision is provided.

Responses:

We really appreciate the reviewer's positive comments and valuable suggestions. We will treasure this precious opportunity and try our best to address the reviewer's concerns point-by-point so that this work can satisfy the high level of *Nature Communications*.

Original Comment 1:

One of my major concern is the DFT calculation: majority of the DFT study is dedicated to comparing the energy profile for the CO-CO vs. COH-CO vs. CHO-CO pathways on the Cu(0) vs. Cu(I) vs. Cu(II) surface. However, all three of the reaction pathways are a dimerization step, which all lead to C₂₊ products. And one of the major experimental claim is that the Cu(II) surface leads to high C₂₊ yield, instead of C₁. The DFT study should've been conducted to compare the energy profile of C₁ vs. C₂₊ reaction pathways on the Cu(0) vs. Cu(I) vs. Cu(II) surface.

Responses:

We thank the reviewer for his/her valuable comments. As the reviewer said, the

energy profile of C_1 vs. C_{2+} reaction pathways is important to the selectivity of the catalyst. Therefore, we further compared the energy profile of the common path reported for C_{2+} products with the C_1 reaction pathway (desorption of CO and hydrogenation of CHO to HCHO), as shown in Fig. R1. On Cu^0/Cu^{2+} , we can find that the direct desorption of CO from Cu^{2+} sites requires 0.82 eV, while further hydrogenation of CO (formed $*CHO$) requires only 0.38 eV; the subsequent C-C coupling process overcomes a low energy barrier (E_a) of only 0.46 eV, which is much less than the further hydrogenation of CHO to HCHO ($E_a = 1.37$ eV). On Cu^0/Cu^{1+} , the direct desorption of CO on Cu^{1+} sites requires 1.42 eV; although the CO hydrogenation to CHO is relatively easy, the further CHO hydrogenation to HCHO needs to cost energy of 1.51 eV. It is obvious that both the CO desorption and CHO further hydrogenation are more difficult to occur compared with the most favored C-C coupling step on Cu^0/Cu^{1+} ($*CO + \#CO \rightarrow O*CC\#O$) and Cu^0/Cu^{2+} ($*CO + \#CHO \rightarrow O*CC\#HO$).

From the above results, it can be expected that for both Cu^0/Cu^{1+} and Cu^0/Cu^{2+} , the selectivity for the C_{2+} products could be more favorable than that for C_1 products. Moreover, Cu^0/Cu^{2+} shows better acceleration for the C-C coupling with OC-CHO than Cu^0/Cu^{1+} with OC-CO, demonstrating the advantages of the Cu^0/Cu^{2+} interface.

Fig. R1. Competitive reaction of C_1 and C_{2+} paths on (a) Cu^0/Cu^{1+} and (b) Cu^0/Cu^{2+} .

Original Comment 2:

The authors used a P-doped synthetic strategy to stabilize a $Cu(0)/Cu(II)$ surface. But there is not even a single mention of the catalytic effect of P during CO_2 electrolysis. I can't help but think is the CO_2RR performance enhanced due to $Cu(II)$ or the influence

of the phosphorous element? It would be important for the authors to provide some experimental/computational data and discussion on this topic.

Responses:

We thank the reviewer for the valuable comments. To verify whether phosphorous also plays a role in the CO₂RR process, we chose Cu₃P as an object to investigate its performance and selectivity toward C₂₊ products. As shown in Fig. R2, the CO-CHO coupling is still the easiest process within the three collaborative pathways, and it needs to overcome an energy barrier of 0.72 eV, which is much higher than that on Cu⁰/Cu²⁺ interface (0.46 eV). Therefore, the enhanced CO₂RR performance could mainly result from Cu²⁺ instead of the phosphorous element. We also referenced previous work. Dismukes, G. C. et al. have reported the catalytic activity and mechanism of Cu₃P for the electrochemical reduction of CO₂ to formic acid [Dismukes, G. C. et al. *Electrochimica Acta*, 391, 138889 (2021)]¹. They proposed that the formation of a surface hydride at isolated *H-CuP₃ sites is the catalytic site for forming both H₂ and formate, and the stronger bonded hydrides (*H-CuP₃ on Cu₃P) favor the production of formate over all other carbon products.

Fig. R2. The energy profile of the three general collaborative pathways (i.e., different coupling paths to C₂₊ products) on Cu₃P.

According to our exploration of phosphorous elements, we verified that phosphorous elements can stabilize the O atoms in Cu-based catalysts, protecting the existence of Cu²⁺ sites. Specifically, we calculated the formation energy of oxygen

vacancy ($E_f(O_{vac})$) on $Cu_3(PO_4)_2$ (Cu^{2+}), CuO (Cu^{2+}), and Cu_2O (Cu^{1+}) respectively, illustrating the stabilization of O elements:

$$E_f(O_{vac}) = E_{sur} - E_{vac} - (E_{H_2O} - E_{H_2})$$

where E_{sur} and E_{vac} are the calculated optimal energies of the complete surface and the surface with oxygen vacancy. E_{H_2O} and E_{H_2} are the small molecule energies of H_2O and H_2 . As shown in Fig. R3, both CuO (-0.08 eV) and Cu_2O (-0.75 eV) have negative formation energies of O_{vac} , implying that the lattice O on their surfaces is easy to dissolve; thus, the positive-valence Cu on CuO and Cu_2O could be difficult to keep on. By comparison, the formation energy of the O_{vac} on the surface of $Cu_3(PO_4)_2$ is 0.82 eV, meaning that the O atoms on $Cu_3(PO_4)_2$ can remain relatively stable on the surface. Therefore, it can be expected that the P element could play an important role in stabilizing the O element, limiting the reduction of Cu^{2+} .

Fig. R3. Formation energy of oxygen vacancies on $Cu_3(PO_4)_2$, CuO and Cu_2O .

Original Comment 3:

All the characterization data (XRD, XPS, XAS) have detected some degree of reduction of the Cu(II) during electrolysis. While the reduction of Cu(II) is expected, the authors have provided data for 1 hour post reaction only. I am interested to know if the Cu(II) catalyst is stable for 5h, 10h or 20h? Can the author show a longer post reaction characterization to demonstrate if the Cu(II) would ever get stabilize or fully reduce?

Responses:

We really appreciate the reviewer's constructive suggestions and comments. According to the reviewer's advice, we conducted XRD and XAFS measurements of $\text{Cu}_2\text{P}_2\text{O}_7$ at the applied potential of -1.40 V in CO_2 -saturated 0.1 M KHCO_3 after 10 hours (Fig. R4-5). The XRD analysis shows that $\text{Cu}_2\text{P}_2\text{O}_7$ and metallic Cu coexist after reaction (Fig. R4). As shown in Fig. R5a, the XANES spectra clearly exhibit that $\text{Cu}_2\text{P}_2\text{O}_7$ stays in higher states than metallic Cu(0) after 10 hours of reaction. Fig. R5b shows the EXAFS spectra of $\text{Cu}_2\text{P}_2\text{O}_7$, in which a distinct peak of Cu-O coordination can be detected at about 1.5 Å in the $\text{Cu}_2\text{P}_2\text{O}_7$ sample for 10 hours post-reaction. The XAFS analysis in conjunction with the XRD results clearly proved the coexistence of Cu^{2+} and Cu^0 species as the CO_2RR goes on for at least 10 h.

Fig. R4. XRD pattern of $\text{Cu}_2\text{P}_2\text{O}_7$ coated on carbon paper after CO_2RR at the applied potential of -1.40 V in CO_2 -saturated 0.1 M KHCO_3 for 10 hours.

Fig. R5. (a) Normalized Cu K -edge XANES spectra and (b) Fourier-transformed Cu K -edge EXAFS spectra of $\text{Cu}_2\text{P}_2\text{O}_7$ before and after 10 hour's reaction at the applied potential of -1.40 V in CO_2 -saturated 0.1 M KHCO_3 .

Original Comment 4:

*For the SEIRAS data, the authors assign the peaks at 1180 and 1520 cm⁻¹ to *OCCHO and *OCCHOH respectively, based on Ref 56 [Angew. Chem. Int. Ed. 2022, 61]. Ref 56 is a fully computational study that cites an experimental study [Angew. Chem. Int. Ed. 2017, 56, 3621]. In the experimental study that actually collected FTIR data, the peaks at ca. 1180 and 1520 cm⁻¹ were assigned to C–O–H and C=O stretching of OCCOH. To me, the second study is more reliable since it is actually an experimental study and it is what Ref 56 is based on. The difference in the peaks assignment do affect the conclusion to some degree. I hope the authors can cite the original study and revise the discussion accordingly.*

Responses:

We really appreciate the reviewer's constructive comment. We acknowledge that it is not sufficient to refer only to the computational study, and experimental research is also needed as reliable support. As for the FTIR measurements were performed with D₂O as electrolyte in a 0.1 M LiOH solution in the aforementioned experimental study, which is different from our test conditions; in order to avoid the influence caused by different experimental conditions, we cite a more suitable experimental study [Chen, X.-M. et al. *J. Am. Chem. Soc.* 143, 7242–7246 (2021)]² and reanalyzed the SEIRAS data.

As shown in Fig. R6, the peak at about 1025 cm⁻¹ could be attributed to the nonplanar vibration (O=C–H) of the *CHO intermediate (1031 cm⁻¹ in Ref. R2); and the peaks detected at about 1255 and 1385-1410 cm⁻¹ could be attributed to the C–O stretch and symmetric vibration (vibration of O–C=O) of the *COOH intermediate, respectively (1253 and 1396 cm⁻¹ in Ref. R2). Additionally, the asymmetric vibration of *OCCHO was detected at 1575 cm⁻¹ (Ref. R2) and 1550 cm⁻¹ [Chen, X.-M. et al. *J. Am. Chem. Soc.* 144, 13319–13326 (2022)]³, respectively. We should mention that Philippe Sautet et al. also theoretically calculated the wavenumbers at 1194 cm⁻¹ and 1534 cm⁻¹ for *OCCHO and *OCCHOH (further hydrogenation of *OCCHO in the OC-CHO coupled path), respectively [Sautet, P. et al. *Angew. Chem. Int. Ed.* 61, e202210060 (2022)]⁴.

In order to confirm the band assignments of the IR to OCCHO on $\text{Cu}_3(\text{PO}_4)_2$ catalyst during CO_2RR , with the help of the Phonopy package, we calculated the force constants (frequency calculation in three directions) and born charge for the related structure, and then conducted post-processing through IR-master software, and obtained the diagram of IR to OCCHO at the $\text{Cu}^0/\text{Cu}^{2+}$ ($\text{Cu}/\text{Cu}_3(\text{PO}_4)_2$) interface (Fig. R7). As shown in Fig. R7, a band peak with strong oscillator strength is located at 1189 cm^{-1} , which is close to our experiment result (1180 cm^{-1}) for *OCCHO.

Based on these literature reports and theoretical calculation results, we deem that the bands detected at about 1180 cm^{-1} and 1520 cm^{-1} in Fig. R6 can prove the existence of *OCCHO(H). We are deeply thankful to the reviewer for raising this important point. We have modified this part of the discussion in the revised manuscript (yellow highlight, Fig. 5a on page 31, line 1-10 on page 13).

Fig. R6. a, b In situ SEIRAS differential spectra of the a $\text{Cu}_3(\text{PO}_4)_2$ and b CuO sample in CO_2 purged 0.1 M KHCO_3 electrolyte in real-time condition.

Fig. R7. Infrared spectrum of OCCHO adsorption by $\text{Cu}^0/\text{Cu}^{2+}$ ($\text{Cu}_3(\text{PO}_4)_2$) simulated by DFT

frequency calculation.

Original Comment 5:

The authors used a Cu/PTFE as a substrate during the flow cell testing. Cu/PTFE itself is a very effective catalyst for C₂₊ products during CO₂RR. The authors should provide the FE performance of Cu/PTFE alone as control.

Responses:

We are grateful to the reviewer for this valuable suggestion. According to the reviewer's advice, we provide the FE performance of Cu/PTFE, which exhibits the FE_{C₂₊} of 58.9~77.2% with the applied current density ranging from -100 to -400 mA cm⁻² (Fig. R8). It proves that the Cu₃(PO₄)₂ catalyst with a maximum FE_{C₂₊} above 90% has excellent CO₂-to-C₂₊ conversion performance intrinsically.

We are deeply thankful for the suggested experiments that would improve the content of our manuscript. We have supplemented the FEs of Cu/PTFE (Fig. R8) in the revised manuscript (yellow highlight, line 17-19 on page 12) and revised Supporting Information (Yellow highlight, Supplementary Fig. 21 on page 22).

Fig. R8. FEs for products over Cu/PTFE at various applied current densities in a flow cell reactor with 2.0 M KOH as the electrolyte.

Minor comments

1. There are a number of oxide-derived Cu catalysts reported to-date (the authors have cited in the introduction). Also, there are P-doped [Appl. Surf. Sci. 2021, 544, 148965.], N-doped [Nat. Nanotechnol. 15, 131–137 (2020)], B-doped [Nature Chem

10, 974–980 (2018).] or cation-ion-doped [Nat Catal (2022) DOI:10.1038/s41929-022-00880-6] Cu catalysts that are somewhat similar to the catalysts reported in this study. So, I do find the novelty aspect of this work is lacking a bit. But the data in this work are complete and comprehensive, so this is not a major concern.

Responses:

We really appreciate the reviewer's valuable comments. In our work, we first theoretically predicted and experimentally constructed the in situ formation of Cu⁰ nanoparticles on Cu²⁺ phosphorus oxysalts with rich and stable Cu⁰/Cu²⁺ interfaces to be the efficient electrocatalyst for CO₂ conversion, and the combined results of in situ SEIRAS and DFT calculations revealed that the stable and abundant Cu⁰/Cu²⁺ interfaces can facilitate the pathway of *CHO directly coupling with *CO to form *OCCHO, thus accelerating the CO₂-to-C₂₊ products conversion. Different from the reported studies mentioned by the reviewer above, our work highlights the role of the peculiar Cu⁰/Cu²⁺ interfaces in promoting selectivity toward C₂₊ products, and we believe that this finding will break new ground for the structural design of electrocatalysts and the construction of synergistic active sites to improve the activity and selectivity of valuable C₂₊ products.

2. Cu²⁺ should have a very signature satellite peak in XPS, that peak is almost completely missing in the CuO sample. Can the authors comments on the reason why?

Responses:

We agree with the reviewer's opinion that the satellite peak of Cu²⁺ should be detected in XPS. According to the reviewer's comments, we performed the XPS measurement of the CuO sample again. As shown in Fig. R9, the satellite peak of Cu²⁺ can be observed in the XPS spectra of Cu 2p region of the CuO control sample. Thanks to the reviewer's valuable comments, Supplementary Fig. 10 in the previous version of the supporting information has been revised (Supplementary Fig. 11 on page 12, yellow highlight).

Fig. R9. XPS spectra of Cu 2p region of CuO control sample collected before and after CO₂RR at -1.40 V vs. RHE for 1 h, showing that the Cu species of CuO were largely reduced to metallic Cu⁰ states after CO₂RR.

3. Line 350 and 351, the authors wrote “...high-efficiency CO₂-to-CO₂+ conversion...”. I believe they meant “CO₂-to-C₂+ conversion...”

Responses:

Thanks to the reviewer’s suggestions, the language and details in the work have been carefully proofread, and we have corrected the typos in the revised manuscript (yellow highlight).

Responses to Reviewer #2

General Comment:

This work fuels into the debate on Cu oxidation state during CO₂RR. The authors screened some Cu catalyst that may be able to stabilize Cu²⁺ and found a bunch of Cu-Phosphate catalyst to be used for electrochemical CO₂ reduction (CO₂RR) to multi carbon products. The authors attributed the high performance towards a stable Cu⁰/Cu²⁺ interface, possibly due to the stability of Cu-Phosphate (either Cu₂P₂O₇ or Cu₃(PO)₄).

A key argument/novelty in this work is the detection of persistent Cu²⁺. Several supporting evidence is presented, namely XAS, in-situ Raman, and in-situ SEIRAS. Although interesting and the CO₂RR activity is decent, I think evidence backing the relevance of Cu²⁺ species in catalysis is not strong.

Responses:

We really appreciate the reviewer's valuable comments and suggestions. We will try our best to address the reviewer's comments point-by-point, and we believe that the in situ experimental evidence and scientific interpretation of the catalytic mechanism in our robust existence of Cu⁰/Cu²⁺ interfaces for highly selective CO₂-to-C₂₊ product conversion are now appreciably improved thanks to the reviewer's comments.

Original Comment 1:

Cu₂P₂O₇ use for CO₂RR has been reported before, (doi:10.1002/anie.202114238), and it does show tendency to produce multi carbon products more than oxide (either Cu₂O or CuO) derived Cu.

Responses:

We really appreciate the reviewer's comments. As the reviewer mentioned, Bao et al. have employed Cu₂P₂O₇ as electrocatalysts, which can be electrochemically reduced to metallic Cu with a highly porous structure in the alkaline environment and achieve a FE of 73.6% for C₂₊ products [Bao, X. et al., *Angew. Chem. Int. Ed.*, 61, e202114238 (2022)]⁵. We acknowledge that Cu₂P₂O₇ used for CO₂-to-C₂₊ conversion has been

reported, while the material preparation method and testing conditions are different with us; what is more, the structure-property relationship proposed in our work is innovative and hasn't been reported:

In our work, we first theoretically predicted and experimentally constructed the in situ formation of Cu^0 nanoparticles on Cu^{2+} phosphorus oxysalts ($\text{Cu}/\text{Cu}_2\text{P}_2\text{O}_7$ or $\text{Cu}/\text{Cu}_3(\text{PO}_4)_2$) with rich and stable $\text{Cu}^0/\text{Cu}^{2+}$ interfaces to be the efficient electrocatalyst for highly selective CO_2 -to- C_{2+} product conversion. The operando XAFS analysis in conjunction with the operando Raman results clearly proved the specific coexistence of Cu^{2+} and Cu^0 species in nanometric Cu/CuPO under CO_2RR conditions, and the combined results of in situ SEIRAS and DFT calculations revealed that the stable and abundant $\text{Cu}^0/\text{Cu}^{2+}$ interfaces derived from Cu^{2+} phosphorus oxysalts can facilitate the pathway of $^*\text{CHO}$ directly coupling with $^*\text{CO}$ to form $^*\text{OCCHO}$, thus accelerating the CO_2 -to- C_{2+} products conversion.

Our work highlights the role of the peculiar $\text{Cu}^0/\text{Cu}^{2+}$ interfaces in promoting selectivity toward C_{2+} products, and we believe that this finding will break new ground for the structural design of electrocatalysts and the construction of synergistic active sites to improve the activity and selectivity of valuable C_{2+} products.

Original Comment 2:

It is confusing that the authors lumped both $\text{Cu}_2\text{P}_2\text{O}_7$ or $\text{Cu}_3(\text{PO})_4$ as Cu-PO , yet the investigation of catalytic activity is focused more on $\text{Cu}_3(\text{PO})_4$, and theoretical study uses $\text{Cu}_2\text{P}_2\text{O}_7$ Why is that?

Responses:

We thank the reviewer for this valuable comment. Both $\text{Cu}_2\text{P}_2\text{O}_7$ and $\text{Cu}_3(\text{PO}_4)_2$ are potential materials for the construction of the $\text{Cu}^0/\text{Cu}^{2+}$ interface, according to our theoretical results. The theoretical result shows that $\text{Cu}_2\text{P}_2\text{O}_7$ has relatively higher electrochemical stability (Fig. R10a); thus, $\text{Cu}_2\text{P}_2\text{O}_7$ was selected as an object to conduct in the following theoretical study.

To verify the feasibility of another candidate $\text{Cu}_3(\text{PO}_4)_2$, we further explored the

catalytic activity of Cu/Cu₃(PO₄)₂. As shown in Fig. R10b-d, we can find that the coupling of #CHO and *CO on the Cu⁰/Cu²⁺ interface built by Cu/Cu₃(PO₄)₂ (* and # represent Cu⁰ and Cu²⁺ sites, respectively) is still the most feasible pathway with an energy barrier of 0.43 eV, which is much lower than the CO-CO or CO-COH coupling processes. This theoretical trend on Cu/Cu₃(PO₄)₂ accords with that on Cu/Cu₂P₂O₇. In general, we can draw the conclusion that the CHO-CO coupling process at the Cu⁰/Cu²⁺ interface has an evident superiority over the OC-CO or OC-COH coupling processes, facilitating the formation of C₂₊ products. Thanks to the reviewer's suggestions, related discussions have been added to the revised Supporting Information (Supplementary Fig. 2 on page 3, yellow highlight).

Fig. R10. **a** Electrochemical stability of candidates, where U_{diss} is the dissolution potential of materials, larger and greener circles represent more stable materials (i.e., larger U_{diss}). **b-d** Energy profiles of different C-C coupling processes (* and # represent Cu⁰ and Cu²⁺ sites, respectively), **b** CO-CO, **c** COH-CO, and **d** CHO-CO at the Cu⁰/Cu²⁺ interfaces constructed by Cu₃(PO₄)₂, and the related transition state structures (TS1~TS3) of different C-C coupling processes.

Original Comment 3:

My main issue with this article is that the argument of persistent Cu²⁺ on Cu-PO catalysts, especially whether the said Cu²⁺ species is relevant catalytically is not convincing. Many literatures have used either Cu₂P₂O₇ or Cu₃(PO)₄ and state clearly that reconstruction can easily happen after cathodic potential application (see e.g.

doi:10.1002/anie.202114238 doi:10.1016/j.electacta.2019.05.030). The authors must explain why they expect their Cu-PO catalysts can be stable and not reduced.

Responses:

We really appreciate the reviewer's valuable questions. As the reviewer mentioned, we found several works reported $\text{Cu}_2\text{P}_2\text{O}_7$ and $\text{Cu}_3(\text{PO}_4)_2$ as electrocatalysts [Bao, X. et al. *Angew. Chem.Int. Ed.* 61, e20211423 (2022); Zhu, Y. et al. *Electrochimica. Acta.* 313, 179–188 (2019)]^{5,6}.

Zhu et al. reported a hierarchical $\text{Cu}_3(\text{PO}_4)_2/\text{Cu-BDC}$ nanosheet array supported by Cu foam as an electrocatalyst, which exhibited excellent electrocatalytic performance for OER and HER in 1 M KOH. For sources of excellent HER performance in $\text{Cu}_3(\text{PO}_4)_2/\text{Cu-BDC}$, they highlight that “Cu/CuO_x will be formed from the partially reduced CuO_x in Cu-BDC nanosheets at the cathode. On the interface of the catalyst, the OH generated from H₂O splitting should be tended to combine with a CuO_x site because of strong electrostatic affinity to the partially positively charged Cu_{2p} species. At the same time, an adjacent Cu site would accelerate H adsorption and thus forming the Volmer process, imparting synergistic HER catalytic activity.” After a long-term HER stability test, the post-HER $\text{Cu}_3(\text{PO}_4)_2/\text{Cu-BDC}$ inherited the overall hierarchical nanosheets configuration; meanwhile, the initial Cu-BDC nanosheets were partially oxidized to CuO/Cu(OH)₂.

In our work, we developed $\text{Cu}_3(\text{PO}_4)_2$ as a highly efficient pre-catalyst for CO₂RR, which achieved 69.7% FE for C₂H₄ in the neutral electrolyte, and 90.9% FE_{C₂⁺} with a *j*_{C₂⁺} of over 300 mA cm⁻² in a flow cell using the alkaline electrolyte. The Cu⁰⁺/Cu²⁺ interfaces could be formed from the partially reduced $\text{Cu}_3(\text{PO}_4)_2$ when a suitable negative potential was applied during CO₂RR. It is worth noting that the reconfiguration of the catalyst in CO₂RR is more complicated than that in HER. For example, the CO₂ atmosphere can affect the morphology, exposed crystal surface, and electronic structure of catalyst reconstruction during CO₂RR [Sargent, E. H. et al. *Nat. Catal.* 3, 98–106 (2020); Cui, Y. et al. *Joule* 3, 1927–1936 (2019)]^{7,8}. Moreover, it has been reported that the dynamic chemical states of Cu and the content of surface CuO_x species in CO₂RR are determined by a “seesaw-effect” between the cathodic reduction potentials and the

OH• radical-involving oxidation [Cui, C. et al. *Nat. Commun.* **13**, 3694 (2022)]⁹. All in all, due to the negative electroreduction potential and complex solution environment during the CO₂RR process, it is untoward to reveal the operando active sites and the catalytic mechanism of electrocatalysts, hence limiting the rational design and development of catalysts with high selectivity, activity, and stability. With these in mind, we have performed in situ Raman, XAFS, and SEIRAS characterizations in conjunction with theoretical calculations to demonstrate the robust existence of Cu⁰/Cu²⁺ interfaces derived from Cu₃(PO₄)₂ during CO₂RR. The Cu²⁺ sites are conducive to the formation of *CHO and then facilely coupled with *CO on the Cu⁰ surface to form the *OCCHO intermediate, leading to high-efficiency CO₂-to-C₂₊ conversion performance.

For Cu₂P₂O₇, Bao et al. have reported that it could be electrochemically reduced to metallic Cu under CO₂RR with 1 M KOH as the electrolyte, which achieved a FE of 73.6% for C₂₊ products at an applied j of 350 mA cm⁻². Meanwhile, in our work, we highlighted that Cu₂P₂O₇ could in situ form nanometric Cu/CuPO with rich Cu⁰/Cu²⁺ interfaces in 0.1 M KHCO₃ during CO₂RR, which could obtain a high FE_{C₂H₄} of 64.0% in the neutral medium. Furthermore, transition metal phosphates have been reported to make a positive contribution to structural stability, on account of which the structural flexibility of the (P₂O₇)⁴⁻ group could stabilize the intermediate states of Cu by changing their local positions with ease [Kang, K. et al. *Nat. Commun.* **6**, 8253 (2015)]¹⁰. Theoretically, we have assessed the electrochemical dissolution potential U_{diss} of Cu₂P₂O₇, and a more positive U_{diss} means higher electrochemical stability. As a result, Cu₂P₂O₇ exhibits optimal electrochemical stability in the neutral medium ($U_{\text{diss}} = 0.94$ V, pH = 7); however, the electrochemical stability of Cu₂P₂O₇ decreased significantly with the increase in pH value ($U_{\text{diss}} = 0.045$ V, pH = 13).

Thanks to the reviewer's comments, we have added the related reference into the performance comparison figure and tables in the revised manuscript (yellow highlight, Fig. 4g on page 30) and the revised supporting information (yellow highlight, Supplementary Table 5 on page 28)

I note the evidence included in this manuscript actually pointed clearly to major

reduction to Cu:

a. XRD (Fig 2a, b) shows that, especially for $\text{Cu}_3(\text{PO})_4$, the catalyst turn primarily to Cu.

Responses:

Thanks to the reviewer's comments. It has been reported that the intensity of XRD diffraction peaks is affected by factors such as the degree of crystallinity, grain size, and material microstructure, so it is not reliable to determine the content of different components in the compound [Patience, G. S. et al. *Can. J. Chem. Eng.* 98, 1255–1266 (2020)]¹¹. Overall, our XRD, XPS and HRTEM results, in conjunction with the operando experimental characterizations (XAFS and Raman) can demonstrate the durability of Cu^{2+} during CO_2RR .

b. EXAFS/XANES data shows significant reduction of the $\text{Cu}_2\text{P}_2\text{O}_7$ towards metallic Cu during (and possibly after) reduction. Key features in the near edge of Cu K edge (Fig 3c) shows clear pre-edge feature at lower energy and the oscillation start to resemble Cu^0 more than Cu^{1+} or Cu^{2+} .

Responses:

We really appreciate the reviewer for pointing out this important comment. We have re-analyzed the XANES data carefully and found out that there are some errors in the calibration of standard samples due to the different test batches, which is reflected in the fact that the absorption edge position of the in situ CuO control sample is more negative than the Cu foil (Fig. 3c in the previous version of the manuscript). We have recalibrated the edge position of standard samples, and as shown in Fig. R11, the $\text{Cu}^0/\text{Cu}^{1+}$ characteristic did appear in the XANES of $\text{Cu}_3(\text{PO}_4)_2$ as the reaction progressed, which indicated that the material was gradually reduced during CO_2RR and the coexistence of Cu^{2+} and Cu^0 . However, it is difficult to evaluate the ratio between Cu^0 and Cu^{1+} based on the characteristic position of the shoulder peak. From Fig. R11, it can be seen that the shoulder peak strength of the CuO control sample is relatively close to that of Cu foil, which is much stronger than that of $\text{Cu}_3(\text{PO}_4)_2$, indicating that the content of Cu^0 after the reaction of $\text{Cu}_3(\text{PO}_4)_2$ is much less than that of CuO.

Thanks to the reviewer's comments, Fig. 3c in the previous version of the manuscript has been revised (yellow highlight, Fig. 3c on page30).

Fig. R11. Normalized Cu *K*-edge operando XANES spectra in CuO and Cu₃(PO₄)₂ catalysts during CO₂RR at - 1.45 V in CO₂-saturated 0.1 M KHCO₃.

- c. The FT-XAS data in Fig 3d also shows disappearance of shorter Cu-O bonds, which indicates the destruction of Cu-O trigonal bipyramid in Cu₃(PO₄)₂ and Cu-Cu bonds start to dominate in the operando and post data.

Responses:

As the reviewer commented, we acknowledge that the Cu-O coordination peak strength decreased and the Cu-Cu coordination peak strength increased during the CO₂RR. However, it's worth noting that the Cu-O coordination did not completely disappear until after the reaction (as shown in Fig. R12), indicating that the component of the Cu oxidation species in Cu₃(PO₄)₂ still remained under CO₂RR. Consequently, the specific coexistence of Cu²⁺ and Cu⁰ species in Cu₃(PO₄)₂ electrocatalyst under CO₂RR conditions could improve the selectivity for C₂₊ products.

Fig. R12. The Fourier-transformed Cu *K*-edge EXAFS spectra of $\text{Cu}_3(\text{PO}_4)_2$ catalysts with standard CuO and Cu foil as controls, indicating that the peak intensity of Cu-Cu in Cu foil is significantly higher than that of Cu-O in CuO.

d. Wavelet transform (Fig 3g) of $\text{Cu}_3(\text{PO}_4)_2$ also shows more similar profile with CuO in situ, and Cu foil.

Responses:

Thanks to the reviewer's comment, as shown in the Morlet wavelet transform (Fig. R13), the $\text{Cu}_3(\text{PO}_4)_2$ catalyst exhibits a distinct feature at 5-10 \AA^{-1} besides the WT maximum at 6-8 \AA^{-1} , which represents the coexistence of Cu-O and Cu-Cu bonds, while there is no Cu-O bond observed in CuO during CO_2RR . Notably, for the Cu-Cu peak of Cu foil and the Cu-O peak of CuO, their intrinsic strengths are very different (Fig. R14), which leads to an obvious signal difference in the wavelet; therefore, it is unreasonable to determine the proportion between the components according to the intensity of the wavelet.

Fig. R13. Morlet WT of the k^3 -weighted operando EXAFS data for the $\text{Cu}_3(\text{PO}_4)_2$ and CuO samples with standard Cu foil and CuO powder as controls, suggesting the coexistence of Cu-O and Cu-Cu bond in $\text{Cu}_3(\text{PO}_4)_2$ during CO_2RR .

Fig. R14. The Fourier-transformed Cu K -edge EXAFS spectra of standard CuO and Cu foil, showing that the peak intensity of Cu-Cu in Cu foil is significantly higher than that of Cu-O in CuO.

e. *The authors rely very much on LCF to show that they have “Cu²⁺” residue, but unfortunately the fitting procedure, goodness of fit, and fitting compared to the spectra is not detailed. Where is the fit performed on? Is it in the energy space, k space or R space? It is actually hard to believe that, comparing the amount of Cu²⁺ shown in Figure 3f (approx. 50%), to Figure 3d mid graph (R space), that should actually show predominantly Cu⁰ (marked by strong Cu-Cu). I believe it is more useful to do first (also second if the energy range is sufficient) shell fitting, to get the type of neighbouring atoms and approximate bond length of the first shell. It will be then more apparent whether this can be attributed to reduced Cu⁰ or higher Cu oxidation (in form of oxides or phosphates).*

Responses:

LCF were processed using the ATHENA module implemented in the IFEFFIT software packages. According to the author's suggestion, we provide all the LCF data and spectra to prove that our fitting results are relatively reliable (Table R1 and Fig. R15). Furthermore, we fitted the EXAFS data of $\text{Cu}_3(\text{PO}_4)_2$ during CO_2RR (Fig. R16 and Table R2). The fitting results show that Cu-O coordination and Cu-Cu coordination

exist simultaneously in the $\text{Cu}_3(\text{PO}_4)_2$ catalyst during CO_2RR .

Thanks to the reviewer's comment, we have supplemented this part of discussion and Fig. R15-16 in the revised manuscript (line 20-26 on page 10, yellow highlight) and revised Supporting Information (yellow highlight, Supplementary Fig. 17-18 on page 18-19; Supplementary Table 1 on page 24).

Table R1. Cu^0 , Cu^{1+} , and Cu^{2+} ratios and for the fitting of the linear combination of XANES spectra.

$\text{Cu}_3(\text{PO}_4)_2$		Time (-1.45 V vs. RHE)					
Cu oxidation state	6 min	12 min	18 min	24 min	30 min	post	
Cu^0	0.04	0.06	0.16	0.23	0.28	0.08	
Cu^{1+}	0.01	0.06	0.12	0.14	0.24	0.41	
Cu^{2+}	0.95	0.88	0.72	0.63	0.49	0.51	
R factor	0.000121	0.000479	0.001947	0.002419	0.006839	0.015517	

CuO		Time (-1.45 V vs. RHE)					
Cu oxidation state	6 min	12 min	18 min	24 min	30 min	post	
Cu^0	0.04	0.18	0.62	0.68	0.73	0.69	
Cu^{1+}	0.00	0.20	0.13	0.13	0.13	0.17	
Cu^{2+}	0.96	0.63	0.25	0.19	0.13	0.13	
R factor	0.001546	0.001479	0.000824	0.000797	0.000817	0.000776	

Fig. R15. LCF of XANES at Cu K-edge. $\text{Cu}_3(\text{PO}_4)_2$, Cu_2O and Cu foil are used as references to represent Cu^{2+} , Cu^{1+} and Cu^0 species, respectively. The fitting results show that there are still 49% composed of Cu^{2+} in $\text{Cu}_3(\text{PO}_4)_2$ after electrochemical reduction via CO_2RR for 30 min.

Fig. R16. Fourier-transformed Cu K-edge EXAFS spectrum of $\text{Cu}_3(\text{PO}_4)_2$ during CO_2RR and its fitting result.

Table R2. Cu K-edge EXAFS curves fitting parameters.

Sample	Path	N	$R(\text{Å})$	$\sigma^2(10^{-3} \text{Å}^2)$	$\Delta E_0(\text{eV})$	R -factor
$\text{Cu}_3(\text{PO}_4)_2$ - in situ	Cu-O	2.6	1.96	6.8	-0.3	0.010
	Cu-Cu	4.6	2.56	-6.2	-2.8	

Notes: N , coordination number; R , distance between absorber and backscatter atoms; σ^2 , Debye-Waller factor; ΔE_0 , inner potential correction accounting for the difference in the inner potential between the sample and the reference compound.

f. Overall, only in-situ Raman shows existence of persistent PO_4 bands. Can this be a “spectator” effect, or unreduced catalyst at the edge of the sample (see e.g., 10.1021/ja960468v)? The authors may include more details on how the in-situ Raman is collected.

Responses:

Thanks to the reviewer’s comments. The in situ Raman measurement was

performed on the Raman spectrometer (LabRAM HR) utilizing an excitation laser with a wavelength of 514 nm and a 50× microscope objective with a numerical aperture of 0.5, 10.6 mm. Before the experiments, calibration was carried out based on the peak at 520 cm^{-1} of a silicon wafer standard. To acquire the information about the electroreduction process of $\text{Cu}_3(\text{PO})_4$ and CuO in Fig. 3a, we utilized a spectroelectrochemical flow cell and detect the in situ Raman of the cathode GDE through a quartz window (Fig. R17). The GDEs of $\text{Cu}_3(\text{PO})_4$ and CuO were taken as the working electrodes, which were prepared by the catalyst sprayed on carbon paper. Platinum wire and Ag/AgCl were used as the counter electrode and reference electrode, respectively. During the in situ experiment, a peristaltic pump was used to control the flow rate of CO_2 -saturated 0.1 M KHCO_3 electrolyte at 10 mL/min, while the flow rate of CO_2 was kept at 10 sccm with a mass flow controller. In order to avoid the influence of the “spectral effect” and the unreduced catalyst at the edge of the sample on the test results, the center part of the quartz window was selected for the in situ Raman test.

According to the reviewer’s suggestion, we have added this part of the discussion to the revised manuscript (yellow highlights, page 17).

Fig. R17. Digital photograph of the homemade cell attached to the PerkinElmer spectrum 100 spectrometer for in situ SEIRAS measurement.

g. Figure 4a, shows reduction feature near 0 V vs RHE. This probably indicates Cu reduction. The authors should study the redox behaviour of both $\text{Cu}_2\text{P}_2\text{O}_7$ and $\text{Cu}_3(\text{PO})_4$ to confirm if it is really stable under bias.

Responses:

We really appreciate the reviewer's valuable comment. According to the previous studies, the reduction feature near 0 V vs. RHE might be ascribed to the reduction of surface Cu^{2+} component [Masuda, H. et al. *ACS Catal.* 9, 6305–6319 (2019)]¹². In our work, we conducted repeated LSV tests on the $\text{Cu}_3(\text{PO}_4)_2/\text{GC}$ sample, and found that the reduction peak had completely disappeared in the second LSV curve (Fig. R18), which is supposed that there are some unstable components on the surface oxides of the working electrode. As confirmed by the in situ Raman, in situ XAFS and other characterization results, the metallic Cu^0 formed as the reduction of $\text{Cu}_3(\text{PO}_4)_2$ started under a constant potential of -1.40 V in the CO_2 -saturated 0.1 M KHCO_3 electrolyte, whereas residual $\text{Cu}_3(\text{PO}_4)_2$ remained present in the bulk.

Thanks to the reviewer's comments, to eliminate the effects of unstable surface oxides on the working electrode, we have replaced the LSV curves of Fig. 4a in the revised manuscript (yellow highlight, Fig. 4a on page 30).

Fig. R18. Repeated LSV curves of $\text{Cu}_3(\text{PO}_4)_2/\text{GC}$ catalysts with respect to RHE at a scan rate of 1 mV/s in the CO_2 -saturated 0.1 M KHCO_3 electrolyte.

Original Comment 4:

*It is interesting that *CO is observed in SEIRAS, but not visible in the in-situ Raman (considering IR are absorbed stronger in aqueous solution). I note that features*

of the in-situ SEIRAS is actually similar for both $\text{Cu}_3(\text{PO})_4$ and CuO pre-catalysts. Similar signals are present, albeit weaker can be seen on CuO precatalyst, just not labelled.

Responses:

Thanks to the reviewer's comments. We consider that the different preparation methods of working electrodes in the two kinds of characterization may cause the SEIRAS characterization to be more sensitive to $\ast\text{CO}$ detection than Raman. For in situ Raman measurements, the working electrodes were prepared by the catalysts sprayed on carbon paper, and through a quartz window. While, for the in situ SEIRAS measurements, a thin Au film with a thickness of ~ 10 nm was prepared on a Si prism using electroless deposition, and then the catalyst-coated Si ATR crystal was taken as the working electrode. Owing to the SEIRA effect, the absorption bands of molecules adsorbed on the chemically deposited films were one order of magnitude as large as those observed on smooth Au electrode surfaces [Osawa, M. et al. *Electrochem. Commun.* 4, 973–977 (2002)]¹³.

In response to the questions about in situ SEIRAS raised by reviewer 1 and reviewer 2, we have carried out a detailed analysis of the spectra: as shown in Fig. R19, the in situ SEIRAS differential spectra of $\text{Cu}_3(\text{PO}_4)_2$ and CuO exhibit peaks at about 2065 cm^{-1} , $1385\text{--}1410\text{ cm}^{-1}$ and 1255 cm^{-1} , which are associated with the absorbed $\ast\text{CO}$, the symmetric vibration (vibration of $\text{O}=\text{C}=\text{O}$) and the $\text{C}=\text{O}$ stretch of $\ast\text{COOH}$, respectively [Chen, X.-M. et al., *J. Am. Chem. Soc.* 143, 7242–7246 (2021); Waegle M. M. et al., *ACS Catal.* 8, 7507–7516 (2018)]^{2, 14}. Additionally, the peak detected at about 1025 cm^{-1} in Fig. R19a could be attributed to the nonplanar vibration ($\text{O}=\text{C}-\text{H}$) of $\ast\text{CHO}$; however, there was no obvious peak at $950\text{--}1100\text{ cm}^{-1}$ in Fig. R19b. Moreover, the distinctive 1180 and 1520 cm^{-1} peaks that could demonstrate the existence of the $\ast\text{OCCHO}$ intermediate were only detected in the in situ SEIRAS differential spectra of $\text{Cu}_3(\text{PO}_4)_2$ [Chen, X.-M. et al., *J. Am. Chem. Soc.* 144, 13319–13326 (2021); Sautet, P. et al., *Angew. Chem. Int. Ed.* 61, e202210060 (2022)]^{3, 4}.

We are deeply thankful to the reviewer for raising this important point. We have modified this part of the discussion in the revised manuscript (yellow highlight, Fig. 5a

on page 31, line 1-10 on page 13).

Fig. R19. a, b In situ SEIRAS differential spectra of the **a** $\text{Cu}_3(\text{PO}_4)_2$ and **b** CuO sample in CO_2 purged 0.1 M KHCO_3 electrolyte in real-time condition.

Original Comment 5:

Raman figure x-axis were different for Fig 3a and 3b, making it difficult to compare between CuO and $\text{Cu}_3(\text{PO}_4)_2$ samples. It is also curious that CO bands are absent even on CuO samples at cathodic potential (see e.g., doi:10.1021/acsami.8b08428, doi:10.1021/cs502128q)*

Responses:

We really appreciate the reviewer's valuable comments and suggestions. According to the reviewer's comments, we carried out the in situ Raman measurement utilizing a spectroelectrochemical flow cell to acquire information about reaction intermediates. Furthermore, the Raman x-axis of the $\text{Cu}_3(\text{PO}_4)_2$ and CuO samples was unified. As shown in Fig. R20a, the stretching of the PO_4^{3-} unit is observed at around $1000\text{--}1100\text{ cm}^{-1}$, and the bands around $450\text{--}650\text{ cm}^{-1}$ are attributed to the bending vibration of the PO_4^{3-} unit. Considering that a moderate decrease in Raman spectra is observed in 60 min, we concluded that metallic Cu^0 formed as the reduction of $\text{Cu}_3(\text{PO}_4)_2$ started, whereas residual $\text{Cu}_3(\text{PO}_4)_2$ remained present in the bulk. Noticeably, the atop-adsorbed CO (CO_{atop}) peak, which acts as an important intermediate for the C-C coupling process, is detected at about 2060 cm^{-1} [*ACS Appl. Mater. Inter.* 10, 28572–28581 (2018)]¹⁵. The in situ Raman measurement results suggest that sufficient

coverage of CO* could be built on the Cu/Cu₃(PO₄)₂ catalyst, promoting further dimerization during CO₂RR.

As for the CuO control sample in Fig. R20b, the Raman peaks of CuO disappeared after 30 min, which indicated its entire reduction. Moreover, no obvious *CO peak was detected in the in situ Raman spectra of the CuO sample, which was caused by its lower CO₂ conversion performance: its CO₂RR catalysis yields CO (<8% FE) and CH₄ (<18% FE), as well as C₂H₄ (<8% FE) in -1.20 ~ -1.50 V.

Thanks a lot for the reviewer's advice. We have replaced the in situ Raman spectra of Fig. 3a in the previous version of the manuscript and modified the relevant discussion in the revised manuscript (yellow highlight, page 9-10 and Fig. 3a on page 29).

Fig. R20. In situ Raman spectra of **a** Cu₃(PO₄)₂ and **b** CuO catalysts, respectively, during CO₂RR at -1.40 V with CO₂-saturated 0.1 M KHCO₃ as the electrolyte.

Original Comment 6:

*Theoretical calculation: the authors based their argument leaning towards Cu²⁺ sites, based on the ease of the protonation of *CO to *CHO or *COH, on Cu, Cu₂O*

and CuO sites representing Cu 0, 1+, and 2+ sites (Fig 1a). On higher Cu oxidation sites, the adsorption conformation of CO will be different due to presence of oxygen, and the first protonation step may not be the most difficult step. The authors should look at possibility of the initial CO₂ adsorption step difference between the Cu oxidation sites, and also with the Cu-PO surface.

a. Figure 1d: what is the meaning to the colour and circle size?

b. Reaction coordinate in Fig 1e-g should be compared to the classical Cu(100) surface as baseline

Responses:

We thank the reviewer for his/her valuable comments. As the reviewer mentioned, CO₂ adsorption and activation could be a necessary and key elementary step in CO₂RR. Here, we explored the adsorption and hydrogenation on Cu (Cu⁰ site), Cu₂O (Cu¹⁺ site), CuO (Cu²⁺ site), and Cu⁰/Cu²⁽¹⁾⁺ interfaces, as shown in Fig. R21. Compared to the initial CO₂ adsorption and hydrogenation step, one can see that the CO₂ adsorption/activation is very poor on Cu⁰, Cu¹⁺, and Cu²⁺ sites with low or even positive adsorption energy. The more positive adsorption energy means poorer activation of CO₂, thus leading to further difficult hydrogenation of CO₂ on Cu⁰, Cu¹⁺, and Cu²⁺ sites. Therefore, the activation and hydrogenation of CO₂ could be the rate-determining step in our system. To further verify whether the activation and hydrogenation of CO₂ could be the rate-determining step, we derived the corresponding Tafel slope by using the formula:

$$\eta = -\left(\frac{2.303RT}{\alpha F}\right) \log i_0 + \left(\frac{2.303RT}{\alpha F}\right) \log i$$

where T is temperature, R/F is the gas/Faraday constant, and α is the charge transfer coefficient. The step of $* + \text{CO}_2 + \text{H}^+/\text{e}^- \rightarrow *\text{COOH}$ has the Tafel slope of $RT/0.5F \sim 120$ mV at 298K. In addition, Tafel analysis for electrocatalytic CO₂RR to C₂H₄ on Cu₃(PO₄)₂ catalyst was also performed in experiments. η for Tafel slopes were calculated using the following equation:

$$\eta = E^0 - E_{\text{Ag}/\text{AgCl}} + 0.217 + 0.059\text{pH}$$

where η is the overpotential, E^0 is the reversible potential (calculated using

tabulated free energies), $E_{\text{Ag}/\text{AgCl}}$ is the applied potential, and pH is the measured electrolyte pH. As shown in Fig. R22, Cu/Cu₃(PO₄)₂ catalyst gives rise to a Tafel slope of 122 mV dec⁻¹, which is close to the above result of the theoretical calculation, indicating the activation and hydrogenation of CO₂ is the rate-determining step for CO₂RR in our system. This result is well consistent with other paper reported [Wang, Z. et al., *J. Mater. Chem. A*, 4, 12616 (2016)]¹⁶.

Fig. R21. Energy profiles from CO₂ to CO at **a** Cu (Cu⁰ site), Cu₂O (Cu⁺ site), CuO (Cu²⁺ site), and **b** Cu⁰/Cu²⁺ interfaces.

Fig. R22. Tafel slopes towards C₂H₄ production of Cu₃(PO₄)₂ during CO₂RR.

a. The circle size is related to the stability of the material. The larger U_{diss} means corresponds to the larger circle size, i.e., the better stability; the different colors represent nothing. To avoid misleading readers, we have modified the form of the display. As shown in Fig. R23a, the larger circles with the greener colors represent the better stability of materials (i.e., larger U_{diss}). Related figures and captions have been added to the revised manuscript (Fig. 1d on page 28).

b. We adopted the reviewer's opinion and explored the same reaction path on Cu(100), which is widely believed to be conducive to C_{2+} products [Gao, M.-R. et al. *J. Am. Chem. Soc.* 144, 259 (2022); Gong, J. L. et al. *Angew. Chem. Int. Ed.* 60, 4879 (2021)]^{17, 18}. The classical material widely believed to be conducive to C_{2+} products, and compared it with the Cu^0/Cu^{2+} interface, as shown in Fig. R23b-d. For the three different coupling paths (CO-CO, COH-CO, CHO-CO), Cu(100) is more inclined to CO-CO coupling with an energy barrier of 0.54 eV, while the Cu^0/Cu^{2+} interface only requires 0.46 eV via CHO-CO coupling due to the easy hydrogenation from CO to CHO. Therefore, the Cu^0/Cu^{2+} interface we constructed does play a certain role in promoting the performance of C_{2+} products in CO_2RR . Related discussions and figures have been added to the revised manuscript (Fig. 1 on page 27).

Fig. R23 a Electrochemical stability of candidates, where U_{diss} is the dissolution potential of materials, larger and greener circles represent more stable materials (i.e., larger U_{diss}). **b-d** Energy profiles of different C-C coupling processes **b** CO-CO, **c** COH-CO, and **d** CHO-CO at the Cu^0/Cu^{2+} interfaces and classical Cu(100) surface.

Original Comment 7:

XPS (SI Fig 10-12), please expand the energy range visual to include the satellite, and include LMM scans, and probably construct a Wagner plot to distinguish Cu⁰, Cu¹⁺ and Cu²⁺ species better

Responses:

We really appreciate and have taken the reviewer's suggestion. As shown in Fig. R24, the peaks detected at about 933.2 and 935.5 eV of initial Cu₂P₂O₇ and CuO can be ascribed to the Cu²⁺ component [Tannenbaum, R. et al. *J. Phys. Chem. C* 112, 1101–1108 (2008)]¹⁹. After 1 h's CO₂RR at -1.40 V, the characteristic Cu⁰/Cu¹⁺ (932.5 eV) peak occupies a dominant position in CuO, indicating the surface Cu²⁺ components of the CuO sample are mainly reduced to Cu⁰/Cu¹⁺ (ref. 19), while the Cu₂P₂O₇ sample continues to exhibit the main component of Cu²⁺ species after CO₂RR. Due to the fact that the binding energies of the Cu¹⁺/Cu⁰ states are difficult to distinguish in the Cu 2p region, we further performed a Cu LMM Auger peak analysis (Fig. R25). The peaks at around 567.7 and 570.0 eV demonstrate that both Cu⁰ and Cu¹⁺ are formed in Cu₂P₂O₇ and CuO after CO₂RR [Hwang, Y. J. et al. *J. Am. Chem. Soc.* 140, 8681–8689 (2018)]²⁰. Furthermore, the feature around 570.4 eV confirms the dominant existence of Cu(OH)₂ (ref. 20), indicating the component Cu²⁺ species are preserved in the Cu₂P₂O₇ catalyst after CO₂RR. However, it should not be ignored that the Cu¹⁺/Cu⁰ components could be partially oxidized in XPS measurement due to the presence of slight surface oxidation in the air [Gong, J. et al. *J. Am. Chem. Soc.* 142, 6878–6883 (2020)]²¹. In that regard, to quantitatively distinguish Cu⁰, Cu¹⁺ and Cu²⁺ species in catalysts during CO₂RR, we have performed in situ XAFS measurements and the corresponding LCF analysis. According to the LCF results, when the catalysts had been electrochemically reduced *via* CO₂RR for 30 min, the majority of the CuO sample had been almost entirely reduced to Cu⁰ (14% Cu²⁺, 13% Cu¹⁺, 73% Cu⁰), while Cu²⁺ was still the main component of Cu₃(PO₄)₂ (28% Cu⁰, 23% Cu¹⁺, 49% Cu²⁺).

Thanks to the reviewer's comments, the XPS spectra of the Cu 2p region in the previous version of the supporting information have been revised, and the Cu LMM Auger spectra are supplemented (yellow highlight, Supplementary Fig. 11-12 and on

Fig. R24. XPS spectra of Cu 2p region of $\text{Cu}_2\text{P}_2\text{O}_7$ and CuO control sample collected before and after CO_2RR at -1.40 V vs. RHE for 1 h, showing that the Cu species of $\text{Cu}_2\text{P}_2\text{O}_7$ exhibited the main component of Cu^{2+} , while Cu species of CuO were largely reduced to metallic Cu^0 states after CO_2RR .

Fig. R25. Cu LMM Auger spectra of the $\text{Cu}_2\text{P}_2\text{O}_7$ and CuO control sample measured after 1 h of CO_2RR at -1.40 V vs. RHE .

Responses to Reviewer #3

General Comment:

This is a very interesting paper reporting a high-activity CuPO-based catalyst for electrochemical CO₂ reduction. The authors have made detailed comparisons with CuO and show convincing evidence that the new catalyst is of high activity. I have some comments, which should be considered before publication.

Responses:

We really appreciate the reviewer's positive comments and valuable suggestions. We will treasure this precious opportunity and try our best to address the reviewer's concerns point-by-point so that this work can satisfy the high level of *Nature Communications*.

Original Comment 1:

The band assignments of the IR to OCCHO etc. intermediates appear to be quite arbitrary. Do authors have some DFT frequency results to support this? Why they cannot be other similar C₂₊ species considering that so many different products are available? Some discussions are required.

Responses:

We thank the reviewer for his/her valuable comments. To support the band assignments of the IR to OCCHO theoretically, we used the Vienna Ab initio Simulation Package (VASP) to simulate the IR of OCCHO intermediates that adsorbed at the Cu⁰/Cu²⁺ (Cu₃(PO₄)₂) interface. With the help of *Phonopy* package, we conducted the frequency calculation in three directions and calculated Born charge for the related structure. Then, we conducted post-processing through *IR-master* software and obtained the diagram of IR to OCCHO at the Cu⁰/Cu²⁺ (Cu₃(PO₄)₂) interface. As shown in Fig. R26a, a band peak with strong oscillator strength is located at 1189 cm⁻¹, which is close to our experiment result (1180 cm⁻¹ in Fig. R26b).

Fig. R26. **a** In situ SEIRAS differential spectrum of the $\text{Cu}_3(\text{PO}_4)_2$ in CO_2 purged 0.1 M KHCO_3 electrolyte in real-time condition. **b** Infrared spectra of OCCHO adsorption by $\text{Cu}^0/\text{Cu}^{2+}$ ($\text{Cu}_3(\text{PO}_4)_2$) simulated by DFT frequency calculation.

Original Comment 2:

*It might be worth reporting the Tafel kinetics, which could help to confirm the rate-determining step, especially $\text{CO}+\text{CO}$, $\text{CO}+\text{CHO}$ are concerned (see review in *ACS Catal.*, 2014, 4, 4364).*

Responses:

We thank the reviewer for his/her valuable comments. The Tafel equation is an effective method to analyze reaction mechanisms in experiments, especially to identify the rate-determining step. However, the interpretation of experimental kinetic data is often affected by many uncertainties, so theoretical calculations are needed to assist in the explanation. According to the paper provided by the reviewer [Liu, Z. P. et al. *ACS Catal.* 4, 4364 (2014)]²², we derived the corresponding Tafel slope by using the following formula:

$$\eta = -\left(\frac{2.303RT}{\alpha F}\right) \log i_0 + \left(\frac{2.303RT}{\alpha F}\right) \log i$$

where T is temperature, R/F is the gas/Faraday constant, and α is the charge transfer coefficient. Since the step ($* + \text{CO}_2 + \text{H}^+/\text{e}^- \rightarrow *\text{COOH}$) involves one electron transfer with the α of 0.5 and there is no electron transfer before it, $* + \text{CO}_2 + \text{H}^+/\text{e}^- \rightarrow *\text{COOH}$ has the Tafel slopes of $RT/0.5F \sim 120$ mV at 298K. In addition, Tafel analysis for electrocatalytic CO_2RR to C_2H_4 on $\text{Cu}_3(\text{PO}_4)_2$ catalyst was also performed in

experiments. η for Tafel slopes were calculated using the following equation:

$$\eta = E^0 - E_{\text{Ag/AgCl}} + 0.217 + 0.059\text{pH}$$

where η is the overpotential, E^0 is the reversible potential (calculated using tabulated free energies), $E_{\text{Ag/AgCl}}$ is the applied potential, and pH is the measured electrolyte pH. As shown in Fig. R27, Cu/Cu₃(PO₄)₂ catalyst gives rise to a Tafel slope of 122 mV dec⁻¹, which is close to the result of the theoretical calculation, indicating the activation and hydrogenation of CO₂ is the rate-determining step for CO₂RR in our system. This result is well consistent with other paper reported [Wang, Z. et al., *J. Mater. Chem. A*, 4, 12616 (2016)]¹⁶. Meanwhile, a recent study has reported that the hydrogenation of CO instead of the C-C coupling is supposed to be the RDS in the formation of C₂₊ products in the CORR [Xu, B. et al. *Angew. Chem. Int. Ed.* 61, e202111167 (2022)]²³. Based on the above analysis, it can be known that the rate-determining step is the activation and hydrogenation of CO₂ instead of the C-C coupling step; while the C-C coupling step is also an important process, which determines the selectivity of C₂₊ products. In conclusion, we have verified in the experimental and theoretical analysis that the CO-CHO coupling path is the most favored C-C coupling step on Cu⁰/Cu²⁺, in which Cu²⁺ can facilitate the formation of CHO.

Fig. R27. Tafel slopes towards C₂H₄ production of Cu₃(PO₄)₂ during CO₂RR.

Original Comment 3:

It is not clear whether the solvation model is switched in DFT calculations, which is important for the solution reaction involving protons (see e.g. J. Am. Chem. Soc.,

Responses:

We thank the reviewer for his/her valuable comments. As mentioned by the reviewer, the solvation environment plays a very important role in DFT calculations for electrochemical reactions. Therefore, we introduced an explicit solvation model (one-layer H₂O molecules) in our models. As shown in Fig. R28, the formation of CHO is much easier compared with that of COH on Cu²⁺ sites; more importantly, the CHO intermediate only needs to overcome a very lowest energy of 0.46 eV to couple with *CO at the Cu⁰/Cu²⁺ interface. In general, although the presence of an explicit solvation model makes some contribution to the change in the energy, it doesn't change our theoretical results. The OC-CHO coupling process at the Cu⁰/Cu²⁺ interface has an evident superiority over the OC-CO or OC-COH coupling processes, thus contributing to the C₂₊ products. The relevant calculation methods and results have been added to the revised manuscript (yellow highlight, pages 6 and 15).

Fig. R28. Energy profiles of different C-C coupling processes after considering solvation effect (* and # represent Cu⁰ and Cu²⁺ sites, respectively), a CO-CO, b COH-CO, and c CHO-CO at the Cu⁰/Cu²⁺ interfaces, and the related transition state structures (TS1~TS6) of different C-C coupling processes.

Original Comment 4:

How about the long-term stability of Cu²⁺ in (Cu₃(PO₄)₂)? Could the stability be improved, i.e. without reducing to Cu⁺ and Cu⁰? The important role of PO₄ in protecting Cu²⁺ could also be modeled in DFT and clarified in the text.

Response:

We thank the reviewer for his/her valuable comments. To understand the role of PO₄ in protecting Cu²⁺, we calculated the formation energy of oxygen vacancy ($E_f(O_{vac})$) on Cu₃(PO₄)₂ (Cu²⁺), CuO (Cu²⁺) and Cu₂O (Cu¹⁺) respectively, illustrating the stabilization of O elements:

$$E_f(O_{vac}) = E_{sur} - E_{vac} - (E_{H_2O} - E_{H_2})$$

where E_{sur} and E_{vac} are the calculated optimal energies of the complete surface and the surface with oxygen vacancy. E_{H_2O} and E_{H_2} are the small molecule energies of H₂O and H₂. As shown in Fig. R29, both CuO (-0.08 eV) and Cu₂O (-0.75 eV) have negative formation energies of O_{vac} , implying that the lattice O on their surfaces is easy to dissolve; thus, the positive valence Cu on CuO and Cu₂O could be difficult to keep on. By comparison, the formation energy of the O_{vac} on the surface of Cu₃(PO₄)₂ is 0.82 eV, meaning that the O atoms on Cu₃(PO₄)₂ can remain relatively stable on the surface. Therefore, it can be expected that the P element could play an important role in stabilizing the O element, limiting the reduction of Cu²⁺.

Attributed to the high thermodynamic and electrochemical stability of Cu₃(PO₄)₂, the stable Cu⁰⁺/Cu²⁺ interfaces could be formed from the partially reduced Cu₃(PO₄)₂ when a suitable negative potential is applied during CO₂RR; however, when a particularly high negative potential is applied, or after a considerable period of reaction, we think Cu²⁺ may be completely reduced to Cu⁰.

Fig. R29. Calculated formation energy of oxygen vacancies on Cu₃(PO₄)₂, CuO and Cu₂O.

References

1. Laursen, A. B. et al. CO₂ electro-reduction on Cu₃P: Role of Cu(I) oxidation state and surface facet structure in C₁-formate production and H₂ selectivity. *Electrochimica. Acta*, **391**, 138889 (2021).
2. Qiu, X.-F., Zhu, H.-L., Huang, J.-R., Liao, P.-Q. & Chen, X.-M. Highly selective CO₂ electroreduction to C₂H₄ using a metal–organic framework with dual active sites. *J. Am. Chem. Soc.* **143**, 7242–7246 (2021).
3. Zhu, H.-L., Chen, H.-Y., Han, Y.-X., Zhao, Z.-H., Liao, P.-Q. & Chen, X.-M. A porous π – π stacking framework with dicopper(I) sites and adjacent proton relays for electroreduction of CO₂ to C₂₊ products *J. Am. Chem. Soc.* **144**, 13319–13326 (2022).
4. Wei, Z. & Sautet, P. Improving the accuracy of modelling CO₂ electroreduction on copper using many-body perturbation theory *Angew. Chem. Int. Ed.* **61**, e202210060 (2022).
5. Sang, J. et al. A reconstructed Cu₂P₂O₇ catalyst for selective CO₂ electroreduction to multicarbon products. *Angew. Chem. Int. Ed.* **61**, e202114238 (2022).
6. Rong, J., Qiu, F., Zhang, T., Fang, Y., Xu, J. & Zhu, Y. Self-directed hierarchical Cu₃(PO₄)₂/Cu-BDC nanosheets array based on copper foam as an efficient and durable electrocatalyst for overall water splitting. *Electrochimica. Acta.* **313**, 179–188 (2019).
7. Wang, Y. et al. Catalyst synthesis under CO₂ electroreduction favours faceting and promotes renewable fuels electrosynthesis. *Nat. Catal.* **3**, 98–106 (2020).
8. Wang, H. et al. Self-selective catalyst synthesis for CO₂ reduction. *Joule* **3**, 1927–1936 (2019).
9. Mu, S. et al. Hydroxyl radicals dominate reoxidation of oxide-derived Cu in electrochemical CO₂ reduction. *Nat. Commun.* **13**, 3694 (2022).
10. Kim, H. et al. Coordination tuning of cobalt phosphates towards efficient water oxidation catalyst. *Nat. Commun.* **6**, 8253 (2015).
11. Khan, H., Yerramilli, A. S., Oliveira, A. D., Alford, T. L., Boffito, D. C. & Patience, G. S. Experimental methods in chemical engineering: X-ray diffraction

- spectroscopy—XRD. *Can. J. Chem. Eng.* **98**, 1255–1266 (2020).
12. Iijima, G., Inomata, T., Yamaguchi, H., Ito, M. & Masuda, H. Role of a hydroxide layer on Cu electrodes in electrochemical CO₂ reduction. *ACS Catal.* **9**, 6305–6319 (2019).
 13. Miyake, H., Ye, S. & Osawa, M. Electroless deposition of gold thin films on silicon for surface-enhanced infrared spectroelectrochemistry. *Electrochem. Commun.* **4**, 973–977 (2002).
 14. Gunathunge, C. M., Ovalle, V. J., Li, Y., Janik, M. J. & Waagele, M. M. Existence of an electrochemically inert CO population on Cu electrodes in alkaline pH. *ACS Catal.* **8**, 7507–7516 (2018).
 15. Deng, Y. et al. On the role of sulfur for the selective electrochemical reduction of CO₂ to formate on CuS_x catalysts. *ACS Appl. Mater. Inter.* **10**, 28572–28581 (2018).
 16. Wang, Z. et al. Enhanced electrochemical reduction of CO₂ to CO on Ag electrocatalysts with increased unoccupied density of states. *J. Mater. Chem. A* **4**, 12616–12623 (2016).
 17. Wu, Z.-Z. et al. Identification of Cu(100)/Cu(111) interfaces as superior active sites for CO dimerization during CO₂ electroreduction. *J. Am. Chem. Soc.* **144**, 259–269 (2022).
 18. Zhong, D. et al. Coupling of Cu(100) and (110) facets promotes carbon dioxide conversion to hydrocarbons and alcohols. *Angew. Chem. Int. Ed.* **60**, 4879–4885 (2021).
 19. Platzman, I., Brener, R., Haick, H. & Tannenbaum, R. Oxidation of polycrystalline copper thin films at ambient conditions. *J. Phys. Chem. C* **112**, 1101–1108 (2008).
 20. Lee, S. Y., Jung, H., Kim, N.-K., Oh, H.-S., Min, B. K. & Hwang, Y. J. Mixed copper states in anodized Cu electrocatalyst for stable and selective ethylene production from CO₂ reduction. *J. Am. Chem. Soc.* **140**, 8681–8689 (2018).
 21. Gong, J. et al. Grain-boundary-rich copper for efficient solar-driven electrochemical CO₂ reduction to ethylene and ethanol. *J. Am. Chem. Soc.* **142**, 6878–6883 (2020).
 22. Fang, Y.-H. & Liu, Z.-P. Tafel kinetics of electrocatalytic reactions: from experiment to first-principles. *ACS Catal.* **4**, 4364–4376 (2014).

23. Chang, X. et al. C-C coupling is unlikely to be the rate-determining step in the formation of C₂₊ products in the copper-catalyzed electrochemical reduction of CO. *Angew. Chem. Int. Ed.* **61**, e202111167 (2022).

REVIEWER COMMENTS

Reviewer #1 (Remarks to the Author):

The authors have addressed all my comments and I am in principle happy to support publication. However, in the revision, the authors provided many Figures to address my comments and many of the Figures/data are not added to the MS/Sl. If those Figures/data are necessary to address my questions regarding the original manuscript, then those Figures/data should be necessary for a general reader. Thus, I would like to request the authors to add all the revision Figures to the SI and also provide appropriate discussion relating to the Figures in the MS/Sl.

Reviewer #2 (Remarks to the Author):

Reviewer report NCOMMS-22-48614A

Direct OC-CHO coupling towards highly C2+ products selective electroreduction over stable Cu0/Cu2+ interface

Xin Yu Zhang et al.

Many thanks to the authors for providing good answers to the reviewers' many queries. In my case the revision has clarified many of the doubt. I have some follow up questions in light of the responses, following this, I think the paper may be a good addition to Nat Comm after the queries are addressed.

1. The authors use dissolution potential to gauge whether the catalyst (pre-catalyst) is stable. However, in my opinion, this is probably less useful measure the stability against cathodic reduction, as it is meant for a metric to quantify the resistance to corrosion/ irreversible anodic oxidation event. One point that stood out from the authors responses is that at high pH the Cu-PO species has lower stability (in terms of dissolution potential). As I understand it, at high current density, the pH (especially local pH) will be very high. Will this mean that the phosphates will slowly degrade and (eventually) dissolve?

2. I think a more relevant metric for reductive stability is the O vacancy formation energy, which has been shown by the authors in response to reviewer 1 question 2

3. Related to question 1, have the authors characterised the catalyst after long term test (Figure 4f)? I understand that there are operando and post reduction XANES analysis shown in Supplementary Table 2, but it is for a much shorter timescale. There is also post TEM in Supplementary Figures 8 and 9, but again only after 30 minutes at -1.4 V RHE. Critically, the information about POx anions' fate is missing. Do the phosphates remain, especially after long term experiments like in Figure 4f? Does the catalyst microstructure change, and more agglomeration of Cu happens? If yes, what is the mechanism such that the performance can be stable? I think proper look into this will also reveal whether the catalyst will be truly stable in industrial time scale.

4. Small point, I think supplementary figure 9 may have missing information on the reaction time, which may be interpreted as in-situ measurement.

5. In Figure 3c and 3d: what is the time scale (in terms of reaction time) of the snapshot in "operando" EXAFS trace? Which timescales for CuO-derived and Cu₃(PO₄)₂ samples shown in Figure 3b-c are in the same order with Figure 3e and 3f? I think it will be beneficial to add this information.

6. I think the cell configuration setup for operando EXAFS is also missing, in the SI I can only find setup information for operando Raman.

Reviewer #3 (Remarks to the Author):

I read the revised manuscript. I am happy with the response and the change made in the manuscript. The work can now be accepted for publication.

Responses to Reviewers' Comments

Responses to Reviewer #1

General Comment:

The authors have addressed all my comments and I am in principle happy to support publication. However, in the revision, the authors provided many Figures to address my comments and many of the Figures/data are not added to the MS/SI. If those Figures/data are necessary to address my questions regarding the original manuscript, then those Figures/data should be necessary for a general reader. Thus, I would like to request the authors to add all the revision Figures to the SI and also provide appropriate discussion relating to the Figures in the MS/SI.

Responses:

We really appreciate the reviewer's positive comments and valuable suggestions. We have added these revised Figures to the revised supporting information (Supplementary Figs. 3, 4, 6, 7, and 27), and provided related discussion in the revised manuscript (yellow highlight, pages 6, 7, and 13).

Responses to Reviewer #2

General Comment:

Many thanks to the authors for providing good answers to the reviewers' many queries. In my case the revision has clarified many of the doubt. I have some follow up questions in light of the responses, following this, I think the paper may be a good addition to Nat Comm after the queries are addressed.

Responses:

We really appreciate the reviewer's positive comments and valuable suggestions. We will treasure this precious opportunity and try our best to address the reviewer's concerns point-by-point so that this work can satisfy the high level of *Nature Communications*.

Original Comment 1:

The authors use dissolution potential to gauge whether the catalyst (pre-catalyst) is stable. However, in my opinion, this is probably less useful measure the stability against cathodic reduction, as it is meant for a metric to quantify the resistance to corrosion/ irreversible anodic oxidation event. One point that stood out from the authors responses is that at high pH the Cu-PO species has lower stability (in terms of dissolution potential). As I understand it, at high current density, the pH (especially local pH) will be very high. Will this mean that the phosphates will slowly degrade and (eventually) dissolve?

Responses:

We really appreciate the reviewer's comments. Attributed to the high thermodynamic and electrochemical stability of CuPO, the partially reduced CuPO can form stable $\text{Cu}^{0+}/\text{Cu}^{2+}$ interfaces when a suitable negative potential is applied in the CO_2RR process. However, when a particularly high negative potential is applied in a strongly alkaline environment after a considerable period of reaction, we think Cu^{2+} may be completely reduced to Cu^0 . Overall, our work highlights the role of the

special Cu⁰/Cu²⁺ interface in promoting the selectivity of C₂₊ products, which will open up new areas for the structural design of electrocatalysts and the construction of synergistic active sites to improve the activity and selectivity of valuable C₂₊ products.

Original Comment 2:

I think a more relevant metric for reductive stability is the O vacancy formation energy, which has been shown by the authors in response to reviewer 1 question 2

Responses:

We thank the reviewer for this valuable comment. We agree with the reviewer that the O vacancy formation energy is a relevant metric for reductive stability, and we have added the discussion related to O vacancy formation energy in the revised manuscript (yellow highlight, lines 5-7 on page 6) and the revised supporting information (Page 4, Supplementary Fig. 3).

Original Comment 3:

Related to question 1, have the authors characterised the catalyst after long term test (Figure 4f)? I understand that there are operando and post reduction XANES analysis shown in Supplementary Table 2, but it is for a much shorter timescale. There is also post TEM in Supplementary Figures 8 and 9, but again only after 30 minutes at -1.4 V RHE. Critically, the information about PO_x anions' fate is missing. Do the phosphates remain, especially after long term experiments like in Figure 4f? Does the catalyst microstructure change, and more agglomeration of Cu happens? If yes, what is the mechanism such that the performance can be stable? I think proper look into this will also reveal whether the catalyst will be truly stable in industrial time scale.

Responses:

We really appreciate the reviewer's constructive suggestions and comments. According to the reviewer's advice, we conducted XPS measurements of Cu₃(PO₄)₂/Cu/PTFE at the current density of -200 mA cm⁻² in 1.0 M KOH after 10 hours. As observed in Fig. R1a, the component Cu²⁺ species are preserved in the

$\text{Cu}_3(\text{PO}_4)_2/\text{Cu}/\text{PTFE}$ after 10 hours' CO_2RR at the current density of -200 mA cm^{-2} [*J. Phys. Chem. C* 112, 1101–1108 (2008)]¹. For the spectrum in the P 2p region (Fig. R1b), the existence of PO_4^{3-} at binding energies of 133.7 eV implies that phosphate structures remained in the catalyst after long-term experiments [*Chem. Mater.* 31, 8, 2892–2904 (2019)]². Further evidence is derived from the O 1s spectrum, the peak at the BE of 533.1 eV can be attributed to the symmetric bridging O in P-O-P group, indicating the phosphate group persists during the test (Fig. 1c) [*ACS Appl. Mater. Interfaces* 8, 22534–22544 (2016)]³. The XPS analysis results clearly proved the remains of PO_x anions and the coexistence of Cu^{2+} and Cu^0 species as the CO_2RR goes on for at least 10 h.

Fig. R1 XPS spectra of $\text{Cu}_3(\text{PO}_4)_2/\text{Cu}/\text{PTFE}$ after 10 hours' CO_2RR at the current density of -200 mA cm^{-2} in 1.0 M KOH in **a** Cu 2p, **b** P 2p, and **c** O 1s regions, indicating the Cu^{2+} species and phosphate group persisted during the test.

Original Comment 4:

Small point, I think supplementary figure 9 may have missing information on the reaction time, which may be interpreted as in-situ measurement.

Responses:

Thanks to the reviewer's comments. The reaction time of Supplementary Fig. 9

is 0.5 h, and we have added it to the revised supporting information (yellow highlight, page 14)

Original Comment 5:

In Figure 3c and 3d: what is the time scale (in terms of reaction time) of the snapshot in “operando” EXAFS trace? Which timescales for CuO-derived and Cu₃(PO₄)₂ samples shown in Figure 3b-c are in the same order with Figure 3e and 3f? I think it will be beneficial to add this information.

Responses:

We really appreciate the reviewer’s valuable comments and suggestions. In our in situ XAS measurements, one spectrum can be acquired on average every 1.5 minutes. The in situ test curves of the samples in Figs. 3c-d were taken at about 36 minutes of reaction. We have added this information in the revised manuscript (yellow highlight, page 17).

Original Comment 6:

I think the cell configuration setup for operando EXAFS is also missing, in the SI I can only find setup information for operando Raman.

Responses:

Thanks to the reviewer’s valuable comments. To measure the *in situ* X-ray absorption spectra, a homemade plastic electrolytic cell was constructed as the three-electrode system (Fig. R2. The graphite rod (spectral purity, 3 mm in diameter) and the Ag/AgCl (3.5 M KCl solution) electrode acted as the counter electrode and the reference electrode, respectively. The carbon paper coated with specific catalysts was used for the working electrode, and a gas inlet for purging CO₂ into the electrolyte (0.1 M KHCO₃) was also contained in the cell. A window (1.1 cm × 1.8 cm) was designed on the cell for the working electrode that enabled XAS measurement under a sensitive fluorescence model while the working potentials were applied.

According to the reviewer’s suggestion, we have added this part of the

discussion to the revised manuscript (yellow highlights, page 17).

Fig. R2 Digital photograph of the in situ XAS electrochemical CO₂RR measurement system, in which the fluorescence model is adopted for collecting spectra.

Responses to Reviewer #3

General Comment:

I read the revised manuscript. I am happy with the response and the change made in the manuscript. The work can now be accepted for publication.

References

1. Platzman, I., Brener, R., Haick, H. & Tannenbaum, R. Oxidation of polycrystalline copper thin films at ambient conditions. *J. Phys. Chem. C* **112**, 1101–1108 (2008).
2. Nguyen, D. C., Tran, D. T., Luyen Doan, T. L., Kim, N. H. & Lee, J. H. Constructing $\text{MoP}_x@ \text{MnP}_y$ heteronanoparticle-supported mesoporous N,P-codoped graphene for boosting oxygen reduction and oxygen evolution reaction. *Chem. Mater.* **31**, 2892–2904 (2019).
3. Chang, Y. et al. Coralloid $\text{Co}_2\text{P}_2\text{O}_7$ nanocrystals encapsulated by thin carbon shells for enhanced electrochemical water oxidation. *ACS Appl. Mater. Interfaces* **8**, 22534–22544 (2016).

REVIEWERS' COMMENTS

Reviewer #2 (Remarks to the Author):

Dear Authors,

I have read the revision, and the work can be accepted into publication. Please consider adding the photograph supplied to review (Figure R2) into the supporting information.